# Microbiome convergence enables siderophore-secreting-rhizobacteria to improve iron nutrition and yield of peanut intercropped with maize

Nanqi Wang [1,6], Tianqi Wang [1,6], Yu Chen[2,6], Ming Wang [3] ✉, Qiaofang Lu [1], Kunguang Wang[1], Zhechao Dou[1], Zhiguang Chi[1], Wei Qiu[1], Jing Dai[1], Lei Niu[1], Jianyu Cui[1], Zhong Wei [4] ✉, Fusuo Zhang [1], Rolf Kümmerli [5] & Yuanmei Zuo [1] ✉

Intercropping has the potential to improve plant nutrition as well as crop yield. However, the exact mechanism promoting improved nutrient acquisition and the role the rhizosphere microbiome may play in this process remains poorly understood. Here, we use a peanut/maize intercropping system to investigate the role of root-associated microbiota in iron nutrition in these crops, combining microbiome profiling, strain and substance isolation and functional validation. We find that intercropping increases iron nutrition in peanut but not in maize plants and that the microbiota composition changes and converges between the two plants tested in intercropping experiments. We identify a *Pseudomonas* secreted siderophore, pyoverdine, that improves iron nutrition in glasshouse and field experiments. Our results suggest that the presence of siderophore-secreting *Pseudomonas* in peanut and maize intercropped plays an important role in iron nutrition. These findings could be used to envision future intercropping practices aiming to improve plant nutrition.

One of humanity's greatest challenges is how to sustainably feed a large population[1], especially in countries such as China, comprising almost a fifth of the world population but having limited available arable land. One solution to guarantee food security is to grow at least two crops simultaneously in the same field because intercropping often increases productivity, resource use efficiency, pest and pathology control, and ecological sustainability[2]. Peanut (*Arachis hypogaea* L.) /maize (*Zea mays*) intercropping is common in China as it is more effective and ecologically more sustainable than peanut monocropping, particularly for small land holders[3]. Peanut is an important oilseed legume. China produces 40% and Northern China Plain approximately 20% of the global peanut production[4] (Fig. 1a). However, peanut yield and quality are severely hampered in this area because of severe iron (Fe) deficiency prevailing in the alkaline and

[1]College of Resources and Environmental Sciences, State Key Laboratory of Nutrient Use and Management (SKL-NUM), National Academy of Agriculture Green Development, China Agricultural University, 100193 Beijing, China. [2]Jiangsu Key Laboratory for the Research and Utilization of Plant Resources, Jiangsu Province Engineering Research Center of Eco-cultivation and High-value Utilization of Chinese Medicinal Materials, Institute of Botany, Jiangsu Province and Chinese Academy of Sciences, 210014 Nanjing, Jiangsu, China. [3]Department of Plant Pathology, The Key Laboratory of Plant Immunity, Nanjing Agricultural University, Nanjing 210095, China. [4]Jiangsu provincial key lab for solid organic waste utilization, Key lab of organic-based fertilizers of China, Jiangsu Collaborative Innovation Center for Solid Organic Wastes, Educational Ministry Engineering Center of Resource-saving fertilizers, Nanjing Agricultural University, Nanjing 210095, China. [5]Department of Quantitative Biomedicine, University of Zurich, Zurich, Switzerland. [6]These authors contributed equally: Nanqi Wang, Tianqi Wang, Yu Chen. ✉e-mail: mwang@njau.edu.cn; weizhong@njau.edu.cn; zuoym@cau.edu.cn

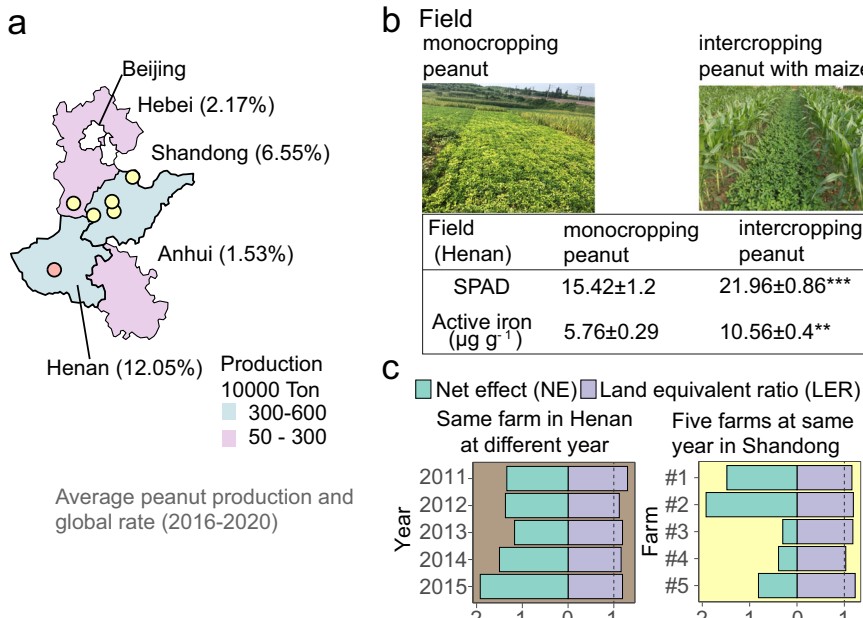

**Fig. 1 | Maize/peanut intercropping improves peanut iron nutrition and yield in calcareous soils. a** Average peanut production and the global rate of four provinces from Northern China Plain and position of field experiments (indicated by dots). **b** Pictures of monocropping and intercropping peanut in field sites and two associated phenotypes of iron-deficiency chlorosis: Soil and Plant Analyzer Development value (SPAD value indicating chlorophyll content) and active iron (indicating iron-deficiency extent) in young leaves. Values are means ± standard deviation (SD) ($n = 4$ biologically independent samples, $p$-value for SPAD: 6.8e-08, $p$-value for active iron: 0.002). Skewed data distributions or data with heterogeneity of variances were transformed using the BoxCox algorithm to meet the assumptions of parametric tests. If data did not meet the assumptions after transformation, non-parametric tests such as the Wilcoxon test were used. The BH algorithm was used to correct $p$-values for multiple comparisons. Asterisks indicate significant differences between two groups: *$p < 0.05$; **$p < 0.01$; ***$p < 0.001$, n.s not significant. Two-sided tests were used for alternative hypothesis testing. **c** Land equivalent ratio (LER) > 1 (LER is defined as the difference between the observed and the expected yield, indicating the land-use efficiency) and net effect (NE) > 0 (NE is defined as the sum of partial relative yields per species, indicating the net effect of intercropping) of peanut/maize intercropping indicate significant advantage in Henan during 2011–2015 and in five farms from Northern China Plain in 2015 under field conditions. Source data are provided.

calcareous soils, making iron insoluble and bio-unavailable[5]. This problem is alleviated through intercropping with maize, which improves the iron nutrition and photosynthetic efficiency of peanut (Fig. 1b) and thereby increases yield and land-use efficiency[5] (Fig. 1c). This intercropping is thought to offer an effective and sustainable way to ensure iron and zinc biofortification to prevent hidden hunger[3].

Here, we use peanut/maize intercropping as a model system to elucidate the exact mechanism leading to increased iron nutrition, particularly focusing on the yet unexplored role of the rhizosphere microbiome. It is known that peanut/maize intercropping improves iron nutrition via direct belowground facilitation through root exudates[5,6]. Higher plants show two alternative iron-deficiency-induced molecular and physiological responses[7,8]. Peanut follows strategy I (dicots and non-graminaceous monocots) and directly reduces Fe(III) to Fe(II) followed by Fe(II) absorption[6,9]. In contrast, maize follows strategy II (graminaceous monocots) and secretes phytosiderophores of the mugineic acid family (MAs) to chelate insoluble Fe(III) followed by MAs-Fe(III) complex absorption[6,10]. Maize outperforms peanut in iron acquisition in alkaline soils because strategy II is less sensitive to high pH[7]. It was proposed that intercropping benefits peanut because it enhances deoxymugineic acid (DMA) secretion from maize root, which solubilizes more Fe(III) in the soil that is then absorbed as DMA-Fe(III) complex by nearby peanut plants[6].

This current plant-plant interaction model for iron utilization does not integrate root-microbiome interactions. This is surprising because it is well established that the rhizosphere microbiome is critical for both plant fitness and iron acquisition[11,12]. Specifically, strategy I and II plants secrete different root exudates under iron-limited conditions, to recruit various rhizobacteria that improve plant nutrition in general and iron acquisition in particular[13-16]. A common mechanism of iron acquisition in these beneficial bacteria operates through the secretion of siderophores to dissolve insoluble Fe(III)[11,17]. For these reasons, we hypothesize that peanut/maize intercropping could modulate the rhizosphere microbiome through the proximity of plant roots, which in turn could promote the mixing of root exudates, microbiome members, their siderophores and thus improve iron acquisition opportunities.

To test our hypothesis, we first examined whether increased iron acquisition in intercropping depends on the presence of a functional rhizosphere microbiome. Subsequently, we combined microbiome profiling, functional strain characterization, substance identification, and greenhouse and field experiments to establish the mechanistic links between cropping systems, belowground root-microbiome interaction networks and iron acquisition. Our study design allows us to identify microorganisms and microbial metabolites that contribute to the increase in nutrient use efficiency and crop yield by intercropping.

## Results

### The rhizosphere microbiome contributes to iron nutrition improvement of peanut intercropped with maize

In the first experiment, we explored whether peanut/maize intercropping is beneficial under greenhouse conditions in naturally iron-limited soils. We found that intercropping indeed improved iron nutrition in peanut (Fig. 2a, b), but not in maize (Supplementary Fig. 1). Specifically, intercropping increased chlorophyll levels (Soil Plant Analysis Development (SPAD) value, 17.6% to 51.5%) and active iron concentration (23.7% to 60.3%, indicating Fe-deficiency extent) in young leaves from 53 days post sowing (dps) onwards (Fig. 2a, b).

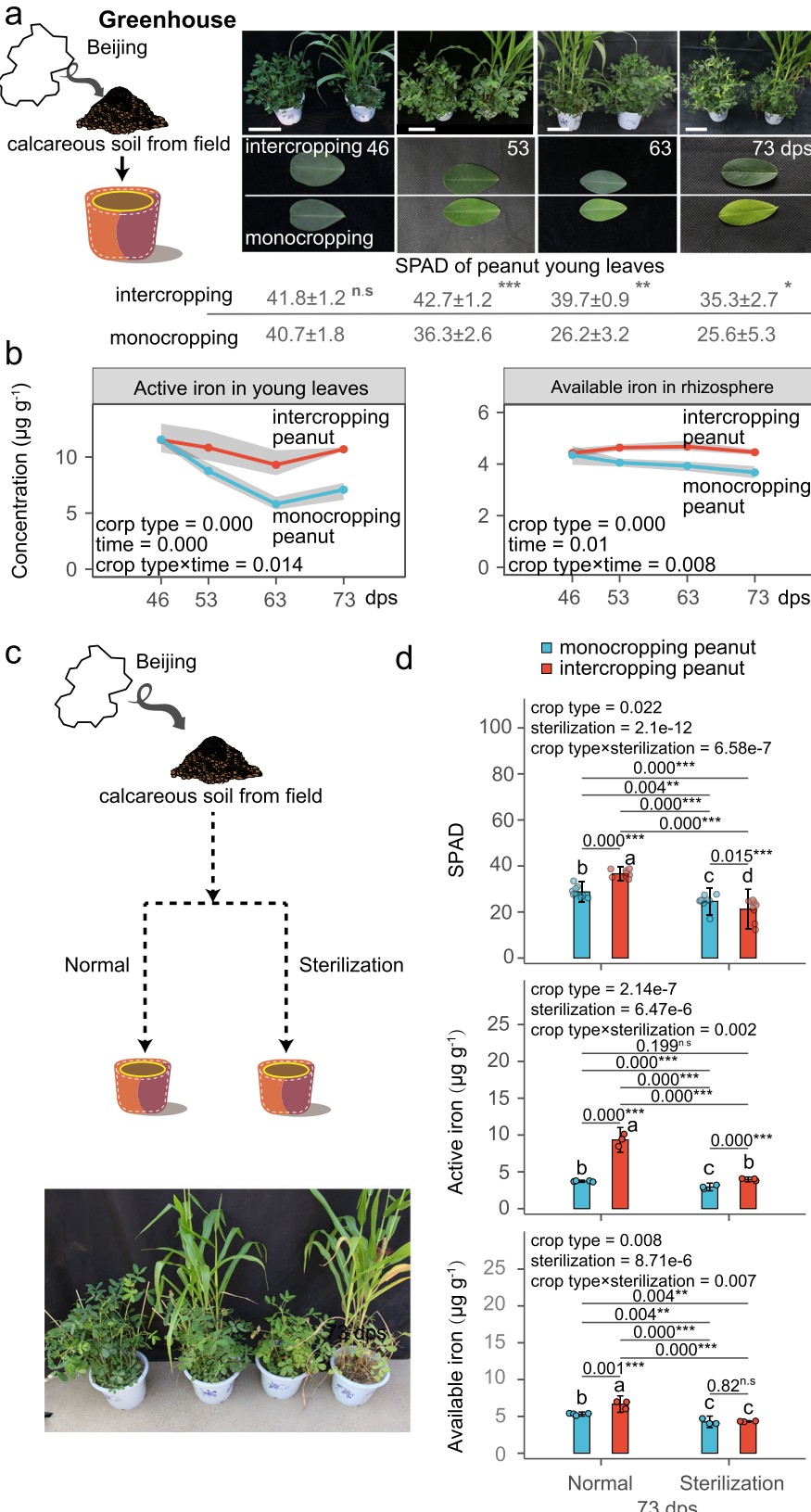

Compared to monocropping peanut, there was a significant increase in the available iron concentration in the rhizosphere (14.3%–21.4%, i.e., iron prevails in a form that can be absorbed by the plant) (Fig. 2b). In contrast, monocropping peanut started to show iron-deficiency chlorosis in young leaves at 53 dps, which progressively worsened over time (Fig. 2a, b).

To test whether the rhizosphere microbiome is involved in the observed iron nutrition improvement in intercropping peanut, we repeated our experiments in normal (unsterilized) and sterilized soils (Fig. 2c). For all the three measured parameters – SPAD, active iron in leaves and available iron in the rhizosphere – there were significant statistical interactions between soil treatments (sterilized *vs.* normal)

**Fig. 2 | Rhizosphere microbiome contributes to the iron nutrition improvement of peanut intercropped with maize. a** Phenotypes and Soil Plant Analysis Development (SPAD) values of plants and young leaves from monocropping peanut and intercropping peanut at 46, 53, 63, and 73 days post sowing (dps). Table shows SPAD values in peanut young leaves. Values show means ± SD ($n = 5$–$8$ biologically independent samples and $p$-values are 0.205, 8.6e-05, 0.002, 0.011 for 46, 53, 63 and 73 dps). Bar indicates 32 cm. **b** Active iron in young leaves and available iron (indicating the amount of iron utilizable by plants) in the rhizosphere of monocropping peanut and intercropping peanut in pots. Points and shaded areas show mean values and 95% confidence interval from four independent biological replicates, respectively. **c** Phenotype of monocropping peanut and intercropping peanut grown in sterile and normal calcareous soil at 73 dps. **d** SPAD values and active iron in young leaves and available iron in the rhizosphere of peanut at 73 dps grown in either normal or sterile soil. Bars with error bars represent mean ± standard deviation (SD) ($n = 10$ biologically independent samples for SPAD value and $n = 3$ for active iron and available iron) and dots represent individual values. For **a**, **b**, and **d** crop type indicates intercropping or monocropping. Parametric Student's $t$-test and ANOVA with LSD as post-hoc test were used for data sets that were normally distributed and had homogeneous variances. Skewed data distributions or data with heterogeneity of variances were transformed using the BoxCox algorithm to meet the assumptions of parametric tests. If data did not meet the assumptions after transformation, non-parametric tests such as the Wilcoxon test, Kruskal–Wallis test with Dunnett T3 test or Scheirer-Ray-Hare test were used. The BH algorithm was used to correct $p$-values for multiple comparisons. Asterisks indicate significant differences between two groups: *$p < 0.05$; **$p < 0.01$; ***$p < 0.001$, n.s not significant. Different letters indicate significant differences among groups. Two-sided tests were used for alternative hypothesis testing. Source data are provided.

and cropping types (intercropping peanut *vs.* monocropping peanut), demonstrating that intercropping peanut was only beneficial and increased iron availability for peanut in the presence of a functional rhizosphere microbiome (Fig. 2d).

### *Pseudomonas* spp. – cross-enriched from maize to peanut – are associated with improved iron nutrition

**Intercropping alters rhizosphere microbiota.** Given that the rhizosphere microbiome is essential for improving iron nutrition in intercropping peanut, we hypothesized that intercropping improves iron acquisition via modification of the peanut microbiome. To test our hypothesis, we used 16S rRNA amplicon sequencing to profile the rhizosphere microbiome of peanut and maize plants from monocropping and intercropping at four-time points (46, 53, 63 and 73 dps, Supplementary Fig. 2). Rhizobacterial communities formed four distinct clusters (monocropping peanut, monocropping maize, intercropping peanut, intercropping maize) at all stages and showed the greatest divergence at 73 dps (Fig. 3a). While bacterial α diversity in the rhizosphere bacterial communities was not affected by intercropping (Supplementary Fig. 3a), we observed that intercropping peanut and intercropping maize microbiomes are more similar than the monocropping peanut and monocropping maize microbiomes (Fig. 3a and Supplementary Fig. 3b). This pattern suggests that maize and peanut microbiome members are exchanged, possibly through the interwoven inter-species root network formed with intercropping (Supplementary Fig. 4). Altogether, our results reveal that intercropping induces both a change in microbiome composition and a convergence between the peanut and maize microbiomes.

**Pseudomonas is the strongest biomarker closely associated with improved iron nutrition.** To detect key bacterial taxa linked with improved iron nutrition, we employed the linear discriminant analysis effect size (LefSe). We identified 10 biomarkers at the genus level (absolute LDA-scores > 3.0 and $p < 0.05$) that were enriched in intercropping peanut compared to monocropping peanut (Supplementary Fig. 5) and were more abundant in monocropping maize compared to monocropping peanut (Fig. 3b). Hence, these taxa are the top candidates for being cross-enriched from maize to peanut in intercropping. We further found that the abundance of those 10 genera positively correlated with either the active iron in young leaves, the iron availability in the rhizosphere, or both metrics (Fig. 3b, Supplementary Fig. 6 showing positive correlation of the two iron metrics). While five genera (*Pseudomonas, Pseudoxanthomonas, Luteolibacter, Allorhizobium-Neorhizobium-Pararhizobium-Rhizobium, Sphingobium*) showed positive correlations with both iron metrics, *Pseudomonas* had the strongest association (Spearman's $\rho = 0.81$) with rhizosphere iron availability (Fig. 3b and Supplementary Data 1). Thus, our analysis reveals five keystone taxa possibly guiding iron nutrition improvement in intercropping peanut, with *Pseudomonas* being the top candidate.

Moreover, we found that intercropping also leads to microbiome shifts in the maize rhizosphere. Particularly, we identified three genera (*Azotobacter, Rubellimicrobium* and *Bryobacter*) that were enriched in intercropping maize compared to monocropping maize (Supplementary Fig. 7).

### *Pseudomonas* rhizobacteria from intercropping show high siderophore-secreting ability

Due to the high capacity of siderophores to solubilize iron[11], we hypothesized that siderophores secreted by rhizobacteria in intercropping could be responsible for the enhanced iron bioavailability in the peanut rhizosphere. To test this hypothesis, we isolated siderophore-secreting rhizobacteria from intercropping peanut using the chrome azurol S (CAS) plate assay (Fig. 4a). We isolated over 300 siderophore-secreting rhizobacteria strains, among which 46 strains had particularly high siderophore-secreting activity (halo diameter on CAS plates > 5 mm, Fig. 4b). Among the top producers, 58.7% were *Pseudomonas* spp. (according to their 16S rRNA sequence), and the majority of these strains (63.0%) had sequence identities > 97% with a representative amplicon sequence variant (ASV487) obtained by the DADA2 algorithm, capturing one particular clade of *Pseudomonas* spp. (Fig. 4b and Supplementary Table 1). Importantly, ASV487 was undetectable in monocropping peanut, but detectable in intercropping peanut and its relative abundance was higher in monocropping maize than intercropping maize (Fig. 4c). Additionally, ASV487-similar strains were isolated in the intercropping peanut rhizosphere at a key stage of the iron nutritional shift (46–53 dps) and at the stage where the iron nutrition gaps between monocropping and intercropping peanut were widest (73 dps) (Supplementary Table 1). These results suggest that ASV487 represents a group of high siderophore-secreting *Pseudomonas* strains that were cross-enriched from maize to peanut during intercropping and are associated with improved iron nutrition in peanut.

Next, we focused on *Pseudomonas* sp. 1502IPR-01, a strain that is 97.7% identical to ASV487 and had the highest siderophore-secreting ability among the isolated strains (Fig. 4b and Supplementary Fig. 8). We used *Pseudomonas* sp. 1502IPR-01 as a representative strain to experimentally explore its role in iron nutrition improvement in intercropping peanut. We sequenced the whole-genome of *Pseudomonas* sp. 1502IPR-01 and found that 1502IPR-01 is similar to *P. extremorientalis* (Fig. 4d and Supplementary Fig. 9). By searching its genome through cluster analysis, we detected the pyoverdine biosynthetic and secretion cluster sharing 51.4–84.1% protein sequence identity with homologous genes of *P. aeruginosa* PAO1, the model strain for pyoverdine research (Fig. 4e and Supplementary Table 2).

To confirm that *Pseudomonas* sp. 1502IPR-01 indeed secretes pyoverdine as its primary siderophore, we isolated the iron-chelating compound (Supplementary Fig. 10) from the supernatant, after cultivation in the iron-limited medium. Our assay revealed that 90.5% ± 0.9% of the iron-chelating ability of supernatants was

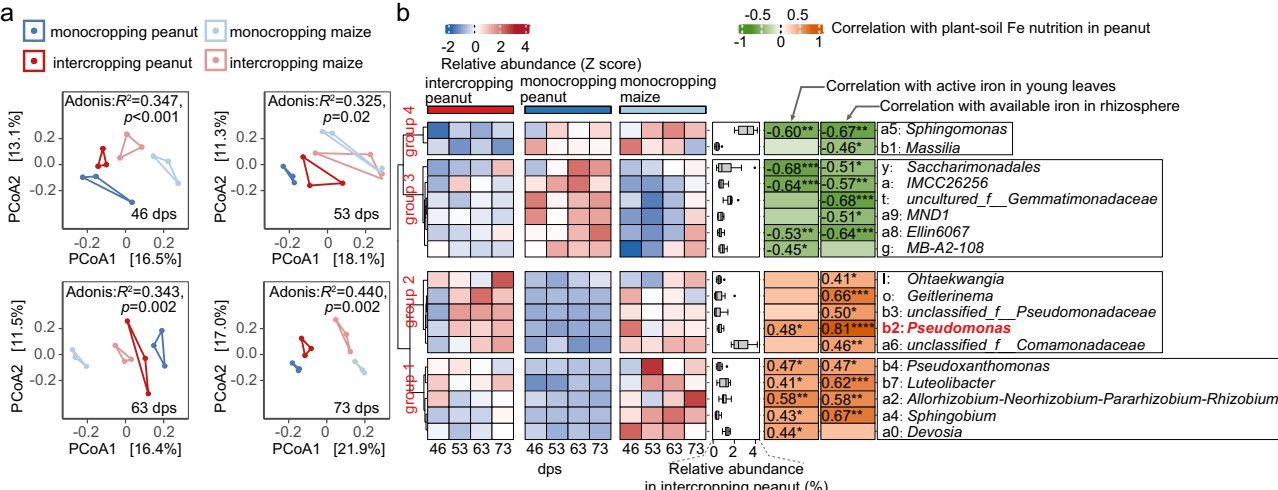

**Fig. 3 | Intercropping induces shifts in rhizobacteria community composition and reveals *Pseudomonas* as one of the top candidate taxa involved in intercropping peanut iron nutrition improvement. a** Unconstrained principal coordinate analysis (PCoA) with UniFrac distances (measuring the distance between communities based on the lineages they contain) of rhizobacteria microbiomes collected from monocropping peanut, intercropping peanut, intercropping maize, and monocropping maize. The results of permutational multivariate analysis of variance (PERMANOVA, also known as Adonis) are shown. For PcoA and Adonis analysis, relative abundances were used. **b** Genus biomarker discriminating monocropping and intercropping peanut, identified by the linear discriminant analysis effect size, LefSe (LDA > 3.0 and *p* < 0.05). Shown are the relative dynamic abundances of the biomarker taxa in intercropping peanut, monocropping peanut and monocropping maize as normalized z-score of mean value of three biologically independent samples and depicted as a heatmap spanning the continuum from low (blue) to high (red) z-scores. Relative abundances of each genus are shown with box plots: center = median, box boundaries = 1st and 3rd quartiles, whiskers = minima and maxima. The table shows Spearman *ρ* correlations between the relative abundances and two metrics describing the plants' iron nutrition state, indicated as heatmap spanning the continuum from negative (green) to positive (pink) correlations. The correlation analyses were based on 24 independent samples. Significant correlations are indicated with asterisks: **p* < 0.05; ***p* < 0.01; ****p* < 0.001; n.s not significant. Exact Spearman *ρ* values with significant levels are listed. The number of biologically independent samples for the heatmap date is three. Two-sided tests were used for alternative hypothesis testing. Source data are provided.

attributable to siderophores (Supplementary Table 3). Structure elucidation and chemical analysis (Fig. 4f, Supplementary Figs. 11–21 and Supplementary Tables 4 and 5, details in "Methods" section) yielded a chemical formula of $C_{57}H_{85}N_{16}O_{25}$, and the chemical structure confirms that the primary siderophore secreted by *Pseudomonas* sp. 1502IPR-01 is a pyoverdine (Fig. 4f).

Next, we assessed the activity of the detected pyoverdine to bind Fe(III) from an insoluble $Fe(OH)_3$ source in suspension. Fe(III) chelation is indicated by a color change from clear to yellow to brown (Fig. 4f). We found that the chelated iron concentration in the solution increased with an equimolar amount of the added pyoverdine, matching the color change of the pyoverdine-Fe(III) control complex (Fig. 4f and Supplementary Fig. 10c). In sum, our results identify pyoverdine as the primary siderophore secreted by *Pseudomonas* sp. 1502IPR-01, and show its high capacity to dissolve Fe(III) from insoluble $Fe(OH)_3$, which corresponds to the main form of iron in calcareous soil[18].

### *Pseudomonas* sp. 1502IPR-01 and its pyoverdine improve peanut iron nutrition in greenhouse and field experiments

To experimentally validate that *Pseudomonas* is a keystone taxon improving iron nutrition through pyoverdine secretion, we conducted greenhouse and field experiments. For the greenhouse experiments, we grew peanut in monocropping and intercropping peanut in sterilized calcareous soil either without treatment as control, or with *Pseudomonas* sp. 1502IPR-01 or its iron-free pyoverdine as treatments. Both treatments significantly increased SPAD values, active iron concentration in young leaves and in the rhizosphere, and improved peanut biomass in monocropping and intercropping peanut (Fig. 5a, b). We observed the same significant improvements when repeating the experiment in normal non-sterile soil in monocropping peanut (Fig. 5b). Thus, *Pseudomonas* sp. 1502IPR-01 and pyoverdine treatments can compensate for iron deficiency in peanut in the absence of a functional microbiome in intercropping and monocropping peanut in normal soil. In contrast, the iron nutrition of maize did not improve when treated with *Pseudomonas* sp. 1502IPR-01 or pyoverdine (Supplementary Fig. 22). However, maize biomass significantly increased with both treatments in sterilized and normal soil in intercropping maize (Supplementary Fig. 22), suggesting that maize benefits from intercropping with peanuts by other means than improved iron nutrition.

Next, we investigated whether the application of *Pseudomonas* sp. 1502IPR-01 and its iron-free pyoverdine are also effective treatments in the field. With intercropping peanut in normal soils, the application of *Pseudomonas* sp. 1502IPR-01 and its iron-free pyoverdine had no effect on iron nutrition and biomass (Supplementary Fig. 23), probably because intercropping itself suffices to mitigate iron deficiency in peanut. Hence, we focused exclusively on monocropping peanut and treated plants with either *Pseudomonas* sp. 1502IPR-01 or pyoverdine (both applied via root watering) at two field sites in North China Plain, characterized by calcareous soils. At both sites, treatments significantly increased SPAD values (increase: 45.9–67.6%), active iron concentration in young leaves (78.2–107.5%), iron availability in the rhizosphere (53.6–73.2%) and crop yield (44.8–89.8%), compared to the control plants that received no treatment (Fig. 5c, d). We further applied a traditional treatment used by local farmers, which involved foliar spraying with EDTA-Fe, a synthetic iron chelator. We observed that treating plants with either *Pseudomonas* sp. 1502IPR-01 or its pyoverdine increased peanut iron nutrition and yield to a similar extent and in one case even more than the traditional EDTA-Fe treatment (Fig. 5d). These results indicate that secreted or added pyoverdine of *Pseudomonas* sp. 1502IPR-01 could effectively chelate Fe(III) in calcareous soils in the field, and this mechanism is responsible for improved peanut iron nutrition and crop yield. Thus, pyoverdine could be considered as an iron fertilizer for sustainable agriculture without the need for external iron supply.

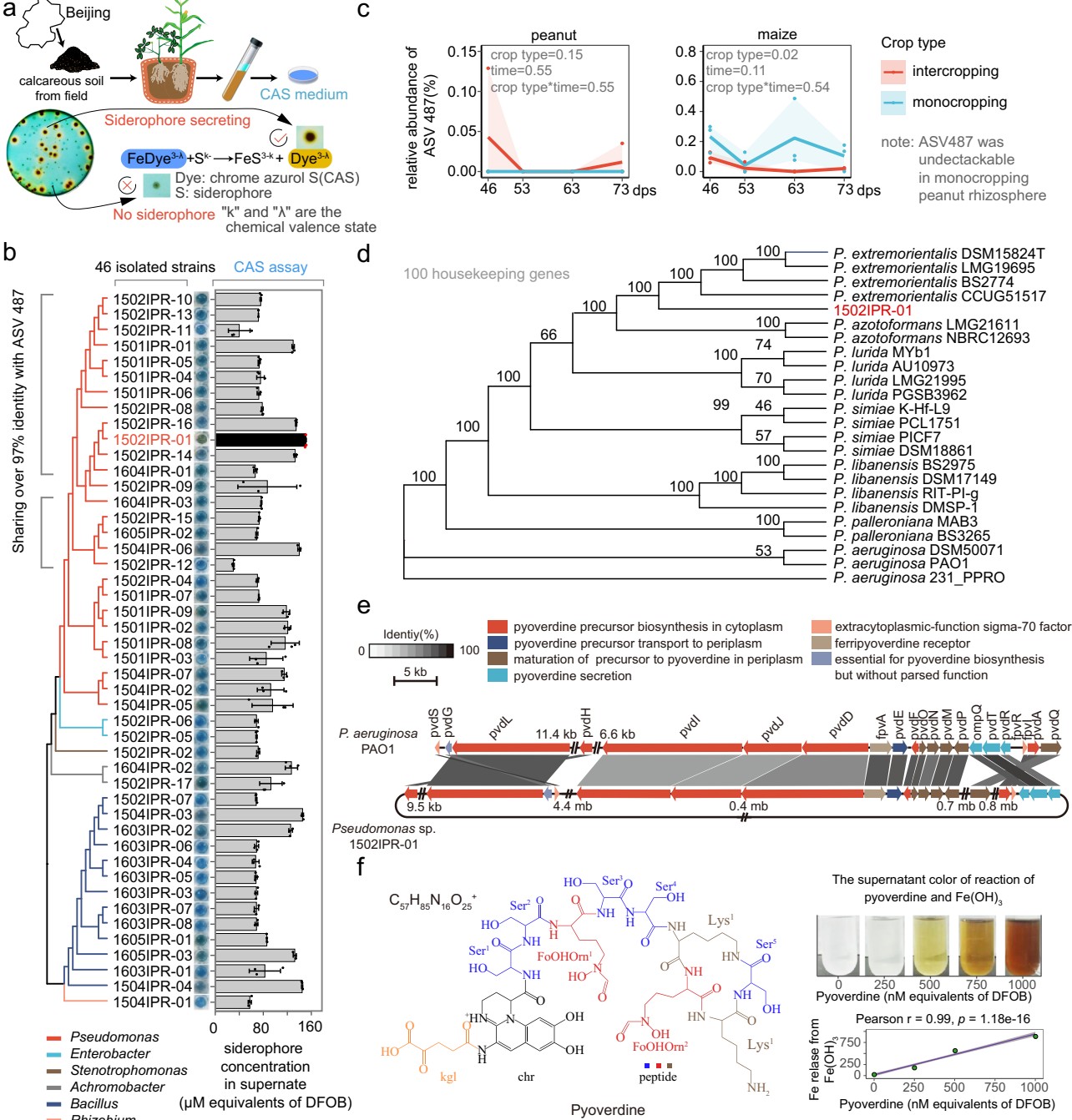

**Fig. 4 | Functional siderophore-secreting rhizobacteria and siderophore characterization. a** Schematic diagram showing the protocol used to isolate siderophore-secreting rhizobacteria. Experimental details are described in the Methods. **b** Phylogenetic tree of the 46 high siderophore-secreting rhizobacteria isolated from intercropping peanut, and the supernatant siderophore concentrations measured under iron-limiting conditions. Bars and error bars represent mean ± standard deviation (SD) ($n = 3$ biologically independent samples) and black dots represent individual values. The photos depict the corresponding CAS assay results. The phylogenetic tree is based on full-length 16S rRNA gene sequences as a phylogenetic marker. Line colors indicate the genus of the isolates. Isolates whose 338F and 806R fraction of the 16S rRNA gene sequence have over 97.0% identity with ASV487 are marked with gray brackets. **c** The relative abundance of ASV487 in monocropping and intercropping peanut and maize over time. Two ways tests were performed with the Scheirer-Ray-Hare test. Points and shaded areas show mean values and 95% confidence interval from four independent biological replicates, respectively. **d** Phylogenetic association of the intercropping peanut rhizosphere isolate *Pseudomonas* sp. 1502IPR-01 with other representatives of *Pseudomonas* spp. strains based on 100 housekeeping genes as phylogenetic markers. The consensus tree is constructed from 1000 bootstrap trees. Numbers on branches show bootstrap supports (%). **e** The pyoverdine biosynthesis secretion and uptake genes of *Pseudomonas* sp. 1502IPR-01. Sequences shared between *Pseudomonas* sp. 1502IPR-01 and *Pseudomonas aeruginosa* PAO1 are connected by bands that are shaded according to their identity level, from black (100.0%) to white (0.0%). The arrow color indicates the function of the genes. **f** The putative chemical structure of pyoverdine from *Pseudomonas* sp. 1502IPR-01 and its capacity to chelate Fe(III) from precipitated Fe(OH)$_3$. The panels show the color change of the supernatant induced by the pyoverdine and the correlation (Pearson correlation coefficient) between the iron release from Fe(OH)$_3$ and the pyoverdine concentration in the reaction mix. The purple line and the gray area indicate a linear regression line and its 95% confidence interval, respectively. The green points indicate mean value across three independent samples. Two-sided tests were used for alternative hypothesis testing. Source data are provided.

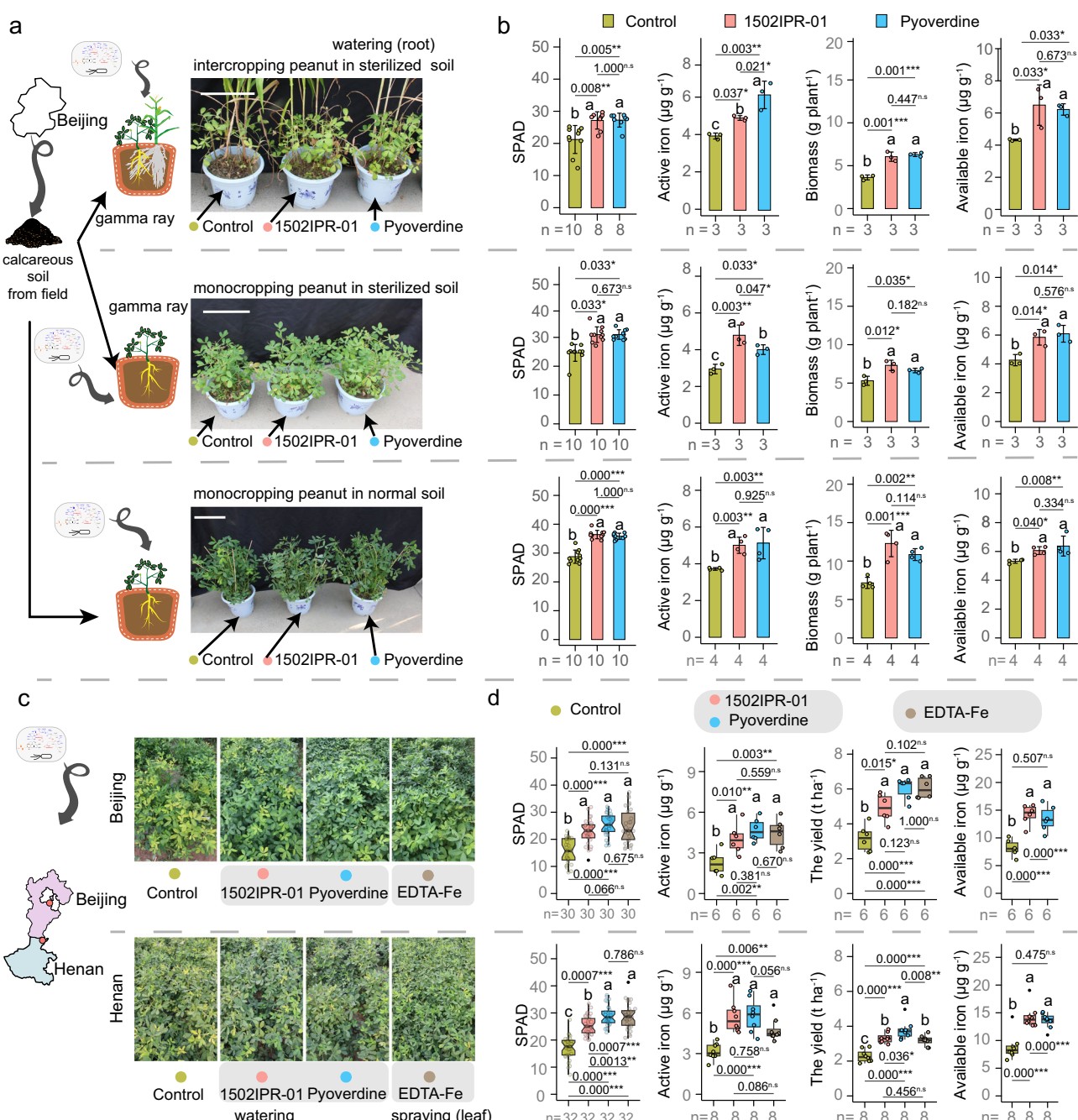

**Fig. 5 | *Pseudomonas* sp. 1502IPR-01 and its siderophore (pyoverdine) prevent iron-deficiency chlorosis and improve peanut growth and yield. a** Phenotypes of peanut plants under control conditions and when treated with *Pseudomonas* sp. 1502IPR-01, unchelated pyoverdine and control group, in intercropping peanut and monocropping peanut in sterilized soil, and monocropping peanut in normal soil in greenhouse conditions (rhizosphere watering application). **b** Soil Plant Analysis Development (SPAD) values and active iron in young leaves, biomass and available iron in the rhizosphere of peanut under the three treatments (control, 1502IPR-01, unchelated pyoverdine of 1502IPR-01) in greenhouse conditions. **c** Phenotypes of peanut plants under control conditions and when treated with either *Pseudomonas* sp. 1502IPR-01 (rhizosphere application), unchelated pyoverdine (rhizosphere application), or EDTA-Fe (foliar spraying) in monocropping peanut in field conditions. **d** SPAD values in young leaves, active iron concentrations in young leaves, available iron in the rhizosphere and peanut yield under the four treatments (control, 1502IPR-01, unchelated pyoverdine, spraying EDTA-Fe) in field conditions.

For **b** bars and error bars represent the mean ± standard deviation (SD) and dots represent individual values. For **d** boxplot center lines show the median, boxplot bounds show the first quartile Q1 and the third quartile Q3, and whiskers show 1.5 (Q3-Q1) below and above Q1 and Q3, and black dot represent outliers and open dots represent individual values, respectively. For **b** and **d** data obeyed normal distribution and homogeneity of variance, ANOVA with LSD as post-hoc test is used in parametric test. For skewness data or data with heterogeneity of variance, data is transformed using BoxCox algorithm for parametric test. Kruskal test with Dunnett T3 test is used if data did not meet the assumptions of parametric test after BoxCox algorithm. Different letters indicate significant differences among groups. Multiple testing corrections are performed by BH algorithm. Asterisks indicate significant differences between two groups: *$p < 0.05$; **$p < 0.01$; ***$p < 0.001$, n.s not significant. For all parameters, $n$ represents the number of biologically independent samples. Two-sided tests were used for alternative hypothesis testing. Source data are provided.

### Validation experiments reveal that pyoverdine directly affects plant iron metabolism

To functionally validate that pyoverdine contributes to improved peanut iron nutrition, we conducted experiments with a defined pyoverdine null mutant (Fig. 6a). Both the lab strain *P. aeruginosa* PAO1 and the natural isolate *Pseudomonas* sp. 1502IPR-01 significantly improved monocropping peanut iron nutrition and biomass in normal soil. In stark contrast, treatment with the siderophore-deficient mutant PAO1 Δ*pvdDpchEF* (Fig. 6b) had no beneficial effects on plants, showing that pyoverdine is required to improve peanut iron nutrition.

We then asked whether pyoverdine secretion feeds back on the activity of the plants' own iron acquisition systems. Indeed, we observed that the ferric-chelate reductase (FCR) activity of peanut is significantly upregulated in the presence of the siderophore-deficient mutant (Fig. 6b), but down-regulated in the presence of pyoverdine-producing wildtype strains and when pyoverdine is added (Supplementary Fig. 24). The same responses occurred in maize, which down-regulated its phytosiderophore DMA in the presence of *Pseudomonas* sp. 1502IPR-01 or its pyoverdine (Supplementary Fig. 24). These experiments highlight that pyoverdine directly affects the iron metabolism of plants.

## Discussion

Intercropping is important for sustainable agriculture and food security[19]. However, the precise mechanisms by which intercropping increases yield often remain unclear. In this study, we find that improved iron nutrition is mediated by the bacterial rhizosphere microbiome and arises through microbiome convergence between the two plant species. Specifically, we find that intercropping enables beneficial bacteria such as *Pseudomonas* spp. to translocate from the maize to the peanut rhizosphere, where these bacteria increase the amount of bioavailable iron through the secretion of their siderophore pyoverdine. The iron chelated to pyoverdine increases iron nutrition in peanut plants and negates iron-deficiency chlorosis in leaves (Figs. 4–7). Siderophores (e.g., pyoverdine) have previously been associated with beneficial effects for plants[20–23], and our study now provides evidence linking bacterial pyoverdine production in planta to improved plant iron nutrition (Fig. 6). This insight closes a major knowledge gap in the research field. Crucially, we show that the uncovered microbe-plant interaction is essential to improve iron nutrition, as intercropping in the absence of a functional microbiome is not beneficial (Figs. 2, 5). Our results further show how detailed knowledge of inter-kingdom interaction mechanisms helps to develop ecologically sustainable agriculture and food security.

Two lines of evidence suggest that the microbiome interactions observed in our greenhouse experiments occur in the field. First, the peanut and maize roots can easily connect across at least 60 cm in the field, matching the results from our pots experiments[24]. Furthermore, root connections across similar distances also occur between maize/faba bean and maize/soybean[25,26]. Thus, the rhizosphere microbiome convergence based on root connections between peanut and maize observed in the greenhouse likely also occurs in the field. Second, our field experiments themselves demonstrated that the addition of *Pseudomonas* sp. 1502IPR-01 or its pyoverdine improved peanut iron nutrition, showing that peanut plants can benefit from iron made available by *Pseudomonas* spp. Hence, pyoverdine-mediated iron mobilization during microbiome convergence could be a major component of improved iron nutrition and yield of peanut intercropped with maize.

Our work strengthens the view that *Pseudomonas* is a keystone taxon involved in plant iron nutrition. We show that *Pseudomonas* sp. 1502IPR-01 and its pyoverdine improve peanut iron nutrition in

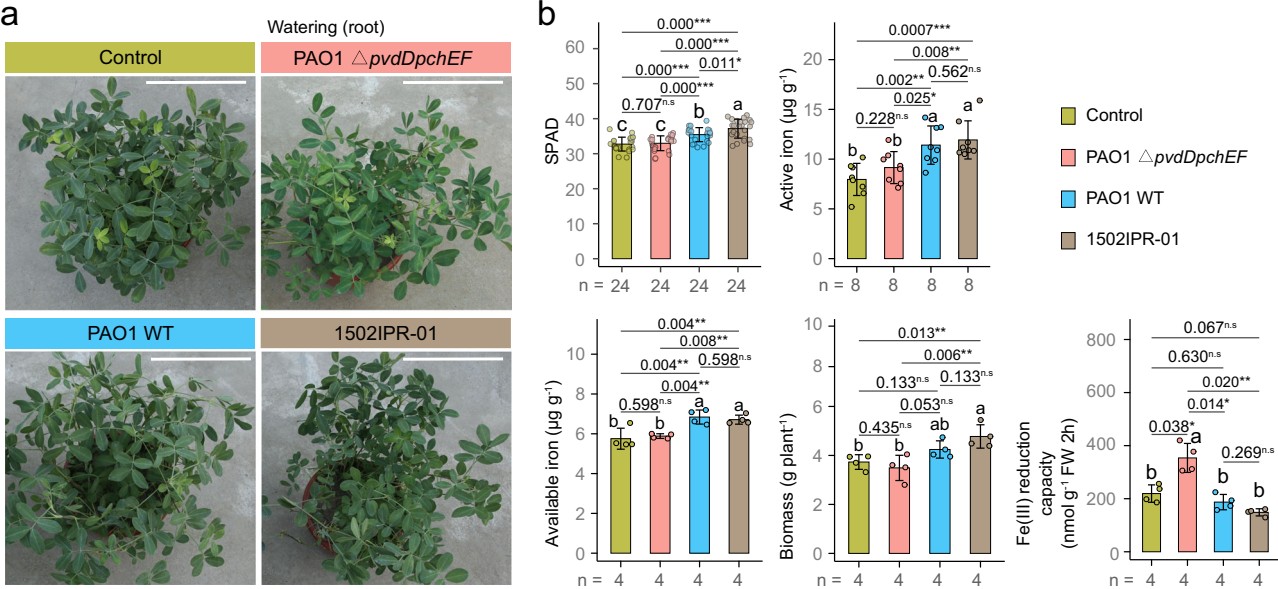

**Fig. 6 | Pyoverdine secreted by *Pseudomonas* spp. is essential to improve peanut iron nutrition and biomass. a** Phenotypes of peanut plants under control conditions and when treated with a *Pseudomonas aeruginosa* PAO1 mutant without pyoverdine secreting ability (PAO1 Δ*pvdDpchEF*), *Pseudomonas aeruginosa* PAO1 wild type (PAO1 WT) with pyoverdine secretion ability, and *Pseudomonas* sp. 1502IPR-01 with pyoverdine secretion ability (all rhizosphere application). **b** Soil Plant Analysis Development (SPAD) values, active iron in young leaves, available iron in the rhizosphere, biomass and Fe(III) reduction capacity of peanut under control conditions and the three treatments (PAO1Δ*pvdDpchEF*, PAO1 WT and 1502IPR-01) in greenhouse conditions. For **b** bars and error bars represent the mean ± standard deviation (SD) and dots represent individual values. ANOVA with LSD post-hoc test was used when data followed the normal distribution and had homogenous variances. For skewed data sets or data with heterogeneous variances, the BoxCox transformation algorithm was applied. When the data still did not meet the assumptions of parametric tests, Kruskal–Wallis test with Dunnett T3 test was used. Different letters indicate significant differences between groups. Multiple testing corrections were performed by the BH algorithm. Asterisks indicate significant differences between two groups: *$p < 0.05$; **$p < 0.01$; ***$p < 0.001$, n.s not significant. For all parameters, n represents the number of biologically independent samples. Two-sided tests were used for alternative hypothesis testing. Source data are provided.

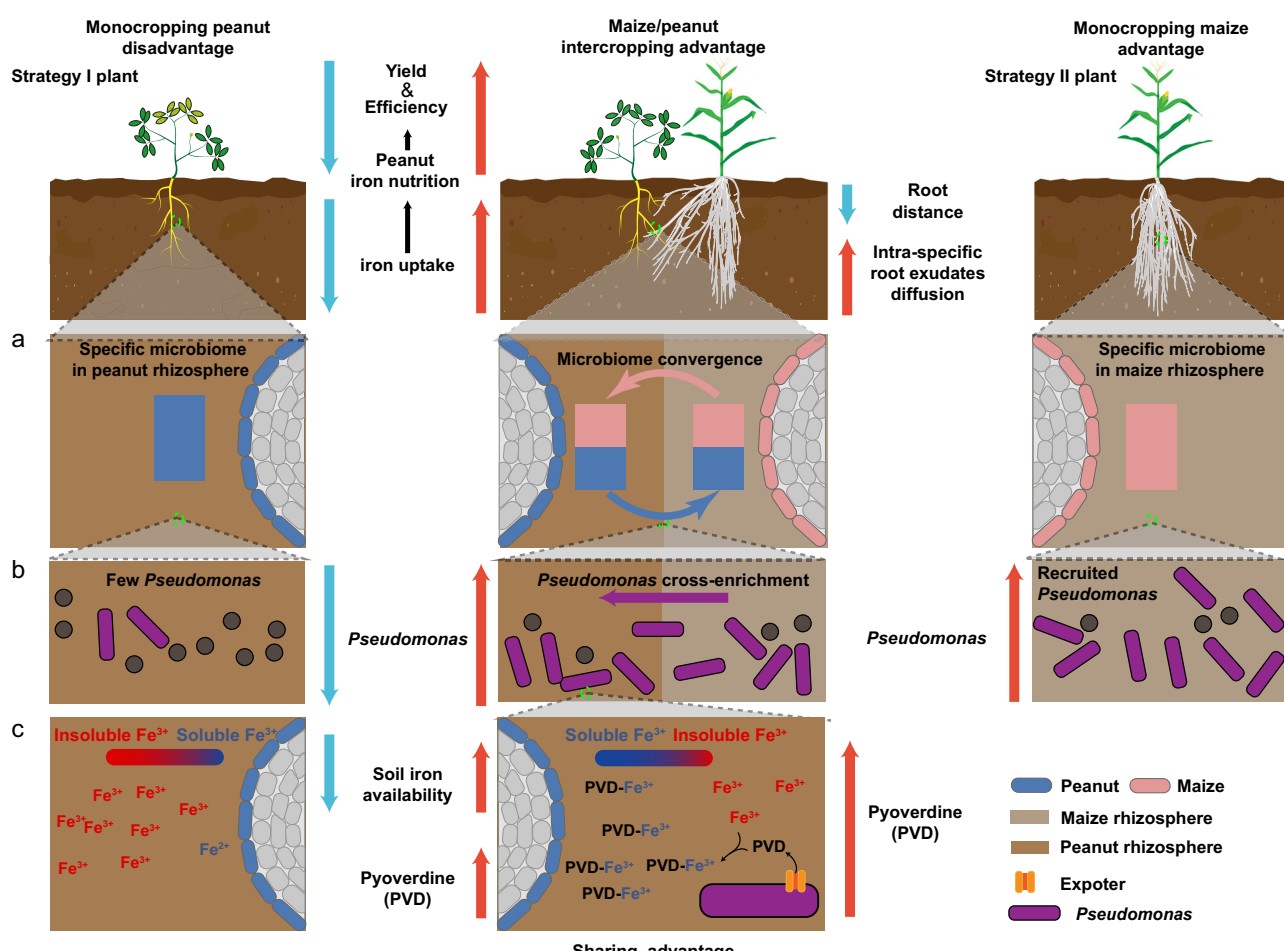

**Fig. 7 | Model of how peanut/maize intercropping enriches for *Pseudomonas* spp. in the rhizosphere to improve iron nutrition via secreting siderophore pyoverdine. a** Rhizosphere microbiome convergence occurs in peanut/maize intercropping through root interactions (whole microbiome level). **b** *Pseudomonas* cross-enriches during intercropping from the maize to the peanut rhizosphere (functional microbe). **c** *Pseudomonas* secretes the siderophore pyoverdine, which increases the iron availability in intercropping peanut rhizospheres (functional microbial metabolite).

calcareous soils in both greenhouse and field conditions through microbiome convergence between peanut and maize (Figs. 3–6). Similar findings regarding the enrichment of *Pseudomonas* in peanut intercropped with maize have been reported in acidic soil[27]. However, important to note is that our analysis revealed additional members of the rhizosphere microbiome that were also associated with iron nutrition improvement, for example, members of the genera *Sphingobium* and *Luteolibacter*. Therefore, it would be important to conduct experimental work with characterized strains of these taxa in the future, to assess their role more closely. The observed beneficial effects in intercropping are likely a bacterial community effect with *Pseudomonas* spp. possibly taking the lead in the process (Figs. 3 and 4). Attributing positive effects to a single species is clearly advantageous when aiming at developing bio-fertilizer treatments for agricultural applications.

The positive effects of pyoverdines on iron nutrition suggest that these molecules take on a yet undescribed role for rhizosphere communities under field conditions. Previous research revealed that specific bacterial siderophores, including pyoverdines, can be beneficial for plants because they suppress plant pathogens in the rhizosphere[23]. The proposed mechanism entails that pyoverdines (and other siderophores) are strong iron chelators, and because they are taxon-specific that can lock iron away from pathogens that do not possess the matching receptors, hence acting as 'public bads'[28]. Competition for iron can therefore drive phytopathogen control and crop protection by natural rhizosphere microbiomes[23]. In intercropping,

pyoverdine seems to primarily serve as a 'public good' that solubilizes iron for the plant and possibly also for rhizosphere microbiome members possessing matching receptors[28]. In principle, pyoverdine could exert a double role in intercropping by suppressing phytopathogens (public bad effect) and by improving iron nutrition of plants (public good effect). Important to note is that plants and microbes may also compete for iron. Siderophores like pyoverdines are primarily produced by bacteria to satisfy their own need for iron. But given that siderophores are highly diffusible they might become available and utilizable for plants[29] (Figs. 5–6). The loss of siderophores to plants could exert positive feedback on bacterial siderophore production as compensation. Conversely, microbial siderophores may also feedback on plant iron-uptake systems. For example, microbial siderophores may chelate iron from phytosiderophore-Fe(III) complexes or Fe(III) chelated by other root exudates, rendering these uptake systems less efficient. Thus, it remains unclear whether the observed iron nutrition improvement in intercropping occurs because plants efficiently exploit bacterial siderophores or whether there is a mutualistic interaction between the two interacting partners.

Our study reveals that pyoverdine can dissolve insoluble iron and make it bioavailable for plants (Figs. 4f, 5, 6). Earlier work has also reported the ability of several plant species to obtain iron from iron-loaded pyoverdines[11,20,21]. However, the exact biochemical mechanism of how plants retrieve iron from pyoverdines remains unclear. One option is that plants take up the entire Fe(III)–pyoverdine complex and retrieve iron through the degradation of pyoverdine. The pyoverdine

backbone consists of a peptide chain and could thus be degraded by proteases. Another option is that plants induce the reduction of Fe(III) to Fe(II) through their ferric-chelate reductase (FCR) to then directly absorb Fe(II). While FCR certainly plays a role, we observed that pyoverdine supplementation led to the down-regulated FCR activity in peanuts and DMA production in maize, which represent the regular iron-deficiency responses of these two plant species[8] (Supplementary Fig. 24). These changes indicate that pyoverdine becomes the preferred route of iron acquisition. In line with this hypothesis, we observed that the down-scaling of the plants' own iron-uptake systems goes along with biomass increases for both peanut and maize plants (Fig. 5 and Supplementary Figs. 22–24), suggesting that both plants benefit and save resources by taking up iron chelated to pyoverdines. This form of nutrient cycling mediated by plant-microbiome interactions is destroyed in sterilized soil, resulting in a decreased SPAD value in intercropping peanut, even lower than in monocropping peanut (Figs. 2d and 5b). However, intercropping can still increase the active iron in young leaves of peanut (Figs. 2d and 5b), suggesting that DMA secreted by maize can directly be utilized by intercropping peanut[6] in sterilized soil. Therefore, our findings indicate that multiple iron sources are tapped during intercropping, including siderophores from bacteria, DMA from maize and possibly also flavins or coumarins from peanut root exudates, to mitigate the plants' iron-deficiency. However, due to the large differences in iron-chelating affinity between pyoverdine and root exudates[10,30], the possibility is that plants balance the iron-uptake strategies between themselves systems and pyoverdine-based ones in intercropping, but future research is needed to test this hypothesis.

While we primarily focused on how iron nutrition is improved in peanut when intercropped with maize, we also detected benefits for maize in a microbiota-dependent manner, suggesting that intercropping benefits both plant species. Maize and peanut could engage in a 'giving-receiving feedback' interaction through the convergence of their microbiomes in intercropping. Peanut plants are legumes and host nitrogen-fixing bacteria in their root nodules[31], such that peanut could improve nitrogen acquisition of maize[27,32]. We indeed found evidence for this hypothesis. Specifically, we detected an enrichment of members of the *Allorhizobium-Neorhizobium-Pararhizobium-Rhizobium* genus and *Devosia* genus from maize to peanut (Fig. 3b). Increased abundance of nodule-forming *Allorhizobium-Neorhizobium-Pararhizobium-Rhizobium* and nonrhizobial nodule-inducing *Devosia*[33] bacteria could increase nodule activity and $N_2$ fixation in peanut, which in return could enhance nitrogen acquisition by nearby maize through reduced competition for soil and fertilizer nitrogen between the two plant species[31]. Furthermore, we found *Azotobacter*, a typical plant beneficial non-symbiotic nitrogen-fixing bacterial genus, to be cross-enriched from peanut to maize, possibly also contributing to improved nitrogen nutrition in maize. Our results are in line with previous findings on microbiota-dependent improvement of nitrogen nutrition[31,32,34–36]. Overall, legume/gramineae combinations are promising candidates for intercropping and are therefore the subject of intensive research in soil science and plant nutrition.

Bioavailability of iron is particularly low in alkaline calcareous soils. That is the main reason why we focused on bacterial members of the rhizosphere microbiome, because bacteria are known for producing a diversity of siderophores, with some of them (e.g., pyoverdine) having a high affinity for iron[22,29,30]. However, important to note is that other factors might also play a role in determining iron availability and benefits of intercropping. For instance, soil nutrients like $NH_4^+$ and $NO_3^-$ (the main forms of nitrogen in the rhizosphere) can impact pH and therefore iron availability. Long-term application of $NH_4^+$ decreases rhizosphere pH, while $NO_3^-$ increases rhizosphere pH[37,38]. However, in our intercropping system, the pH remained stable across nutrient stages (Supplementary Fig. 6b), implying that the nitrogen cycle may

have no direct effect on iron nutrition improvement in our system. The reason may be that calcareous soils have strong pH buffering capacity[5]. Moreover, soil properties, like $NH_4^+$, $NO_3^-$, or organic carbon, can potentially affect microbial community structure and function, meaning that beneficial taxa like *Pseudomonas* might not be equally abundant in all soils. Finally, while we focused on bacteria other members of the microbiota might also be involved in intercropping benefits. Top candidates are the nitrogen-fixing rhizobium (as found in our study for maize) and arbuscular mycorrhizal fungi. Understanding the effects of such multi-kingdom plant-bacteria-fungi interactions across soils with different properties and geographical scales for intercropping will be the challenge of future work.

Our results show that intercropping in natural soils is sufficient to improve iron nutrition in calcareous soils (Supplementary Fig. 23). However, intercropping might not always be possible. For those situations, the insights from our study provide a simple practicable treatment, which consists of the addition of *Pseudomonas* sp. 1502IPR-01 or its pyoverdine to the rhizosphere. Our treatment has several advantages over the use of traditional iron fertilizers such as synthetic EDTA-Fe with low efficiency and high environmental risk in soil[39,40]. These issues do not apply to the pyoverdine treatment because it directly solubilizes iron from natural stocks in soil, pyoverdine itself is biodegradable[22,29,30], and *P. extremorientalis* (closest relative of *Pseudomonas* sp. 1502IPR-01) is classified as a harmless biosafety risk 1 organism that is unlikely to cause disease in humans[41]. Hence, *Pseudomonas* sp. 1502IPR-01 and its pyoverdine can be considered ecosystem-friendly biofertilizers. Additionally, pyoverdines can also enhance copper bioavailability and thus have the capacity to improve plant nutrition for other microelements[42]. Given its dual role in microelements (especially iron) fertilization and phytopathogen control[22,23,30,42,43], pyoverdine could simultaneously reduce the use of synthetic microelements-fertilizers and pesticides, which could present a critical step forward toward sustainable agriculture, improved food security and environmental conservation.

In summary, we elucidated a mechanism of how intercropping benefits plants through rhizosphere microbiome convergence. Our study highlights that the benefit of intercropping from belowground facilitation leads to a gained function for peanut through a cross-enrichment of rhizosphere microbiome members obtained from maize (Fig. 7). Our findings offer applied opportunities for a probiotic treatment to improve plant health and crop yield in situations where intercropping is not possible. While we have identified a probiotic strain and its secreted beneficial compound, there is great potential to improve the process through bioengineering. Our mechanistic insights on iron nutrition improvement offer perspectives and should trigger interest in promoting other intercropping systems with other plant combinations to increase both yield and sustainability.

## Methods

### NE and LER calculation

To quantify the benefits of peanut/maize intercropping, we calculated the land equivalency ratio (LER) and the net effect (NE) of this intercropping system for various field sites and years. LER > 1 indicates that intercropping has an advantage over monocropping in land-use efficiency. NE > 0 indicates that intercropping has more absolute yield than monocropping[1].

The NE is defined as the difference between the observed yield and the expected yield.

$$NE = (Y_1 + Y_2) - (EY_1 + EY_2) \qquad (1)$$

where $Y_1$ and $Y_2$ are observed yields of species 1 and 2 in the intercropping, and $EY_1$ and $EY_2$ are expected yields of the two species, calculated as the product of the monocropping yield and the

land share[1].

$$EY_1 = M_1 \times LS_1 \tag{2}$$

$$EY_2 = M_2 \times LS_2 \tag{3}$$

where $M_1$ and $M_2$ are yields per unit area of species 1 and 2 in mono-cropping, and $LS_1$ and $LS_2$ are the land shares of species 1 and 2 in intercropping. The land share was calculated based on the densities of a species in the intercropping and in the monocropping or on the basis of the number of rows or plant arrangements[1].

The **LER** is defined as the sum of partial LERs (relative yields) per species (pLER1 and pLER2):

$$LER = pLER_1 + pLER_2 = \frac{Y_1}{M_1} + \frac{Y_2}{M_2} \tag{4}$$

where $Y_1$ and $Y_2$ are the yields (per unit of the total area of the inter-cropping) of species 1 and 2 in intercropping, and $M_1$ and $M_2$ are the yields of species 1 and 2 in monocropping (same as above)[1].

## Experimental design for peanut/maize intercropping under pot conditions

We conducted intercropping and monocropping experiments in pots. A typical iron-deficient calcareous sandy soil was collected from 0 to 20-cm-depth in Lihua village, Daxing district of Beijing, China (116°15′ E, 39°35′ N). Soil properties before fertilizer application were deter-mined according to protocols from the previous literature[6] and listed in Supplementary Table 6. Each pot contained 8 kg of soil amended with basal fertilizers [composition (mg kg$^{-1}$ soil): N 100 (as Ca(NO$_3$)$_2$·4H$_2$O), P 150 (as KH$_2$PO$_4$), K 100 (as KCl), Mg 50 (as MgSO$_4$·7H$_2$O) and Zn 5 (as ZnSO$_4$·7H$_2$O)].

Peanut (*Arachis hypogaea* L. cv. Luhua14) and maize (*Zea mays* L. cv. Zhengdan958) were used. Three cropping treatments were set up. Six peanut plants or three maize plants were grown in a pot for the monocropping treatments (monocropping peanut and monocropping maize). Three peanut plants with three maize plants were grown in a pot for the intercropping treatments. The plants were grown under natural light conditions with 28–33 °C air temperature, 400–450 mmol m$^{-2}$ s$^{-1}$ light intensity and 70–75% relative humidity. During the experiment, soil water content was kept at approximately 80% of the field capacity. Plant samples of peanut were harvested to assess the nutrition status at 46, 53, 63, and 73 days post sowing (dps). At the same time, pots were destroyed to acquire complete soil. Bulk soil was removed by gently shaking. The roots of intercropping maize and intercropping peanut were carefully separated from each other when shaking. Rhizosphere soil was collected for microbiome and soil properties analysis by removing the soil closely adhered to the roots using a sterilized fine brush (Supplementary Fig. 2). The peanut plants and rhizosphere soils from the same pots were mixed into one repli-cate. Each treatment had four replicates at each time point.

## Design for soil sterilization experiment in pot conditions

To assess the role of the rhizosphere microbiome in intercropping peanut iron nutrition improvement, monocropping peanut and pea-nut/maize intercropping were grown in both normal and sterilized soil. Soil was prepared as described above. For soil sterilization treatment, the soil was γ-irradiated with a maximum dose of 20 kilograys (24 h, $^{60}$Co-γ, Beijing Atomic High-tech Jinhui Radiation Technology Appli-cation Co. Ltd.). Plant species, growth conditions, and crop manage-ment were the same as in the experimental design for peanut/maize intercropping in pot conditions. At 73 dps, plant samples of peanut were harvested to assess the iron nutrition status, and rhizosphere soil was collected to measure available iron concentration. The peanut

plants and rhizosphere soil from the same pot were mixed and treated as one replicate. Each treatment had three or four replicates.

## Measuring iron-related indicators in plants and soil

**SPAD value.** Chlorophyll levels in fresh young leaves (top fully emerged leaves) from peanut was measured using a chlorophyll meter (Konica–Minolta, Osaka, Japan). Each leaf was measured in four repli-cations and averaged while avoiding the main leaf veins.

**Ferric-chelate reductase (FCR) activity of peanut plants.** According to previous literature[44], the fresh peanut plants were washed and then the roots were immersed in saturated CaSO$_4$ solution for 30 min. After washing, peanut plants were transferred to a 100 Ml reagent consisting of 0.4 Mm 2,2′- bipyridine and 0.1 Mm Fe(III)-EDTA, for a 2-h incuba-tion, protected from light. The absorbance of the solution at 523 nm ($A_{523}$) was measured. Ferric-chelate reductase activity was calculated with the formula (5) below and water was used as a control:

$$\text{Ferric} - \text{chelate reductase capacity}(\text{nmol g}^{-1}\text{ FW 2h})$$
$$= A_{523} \times 100/(\text{FW} \times 8650) \times 10^9 \tag{5}$$

where $A_{523}$ is the absorbance of the solution under 523 nm, FW is the fresh weight of the root, and 8650 is the molar absorptivity of Fe(II)-2,2′- bipyridine (L mol$^{-1}$ cm$^{-1}$). 100 is the volume of plant incubation solution.

**Deoxymugineic acid (DMA) secretion capacity of maize plants.** DMA is only phytosiderphore secreted by maize, so we used phytosi-derphore secretion-determination method[45] to determine the DMA secretion capacity. The fresh maize plants were washed and then the roots were immersed in saturated CaSO$_4$ solution for 30 min. After washing, maize plants were transformed to 200 mL water for a 4-h incubation, protected from light. Before incubation, 10 mg L$^{-1}$ micro-pur (Katadyn Products Inc. Wallisellen, Switzerland) was added to prevent microbial degradation of phytosiderphores. 10 mL above-incubated solution was added 0.5 mL of 1 mM FeCl$_3$ and shaken for 15 min, then 1 mL sodium acetate buffer (pH 7.0) was added and the solution was shaken for a further 10 min. To reduce Fe(III) to Fe(II), the solutions were filtered into 0.25 ml of 6 M HCl and then 0.5 mL 80 g L$^{-1}$ hydroxylamine hydrochloride was added. The solution was incubated at 55 °C for 30 min. Then, the solution is mixed with 0.25 mL 2.5 g L$^{-1}$ ferrozine and 1 mL 2.0 M sodium acetate buffer (pH 4.7). After 5-min incubation, the absorbance of the solution at 562 nm was measured. DMA secretion activity was calculated with the formula (6) below by using water as a control:

$$\text{DMA secretion capacity}(\text{nmol g}^{-1}\text{ FW 4h}) = A_{562} \times 1.35 \times 200/$$
$$(\text{FW} \times 1/0.000045) \times 10^9 \tag{6}$$

where $A_{562}$ is the absorbance of the solution under 562 nm. FW is the fresh weight of the root. 1/0.000045 is the molar absorptivity (L mol$^{-1}$ cm$^{-1}$). 200 is the volume of plant incubation solution. 1.35 is the volume ratio of the final and original reaction solution.

**Active iron in young leaves.** Fresh young leaves from peanut were washed with distilled water and collected to assess HCl-extractable iron ('active iron') according to the previous literature[6]. 1 g of young leaves was taken and mixed with 10 mL of 1 M hydrochloric acid. The supernatant was shaken for 5 h at 150 r.p.m. and then filtered with filter paper. The iron concentration was measured by inductively coupled plasma-optical emission spectrometry (ICP-OES) using a 7300DV sys-tem (Perkin Elmer, Waltham, USA). Active iron in young leaves was calculated with the formula (7) below and deionized water was used as

a control:

$$\text{Active iron in young leaves}(\mu g\,g^{-1}) = (CONCN_{Fe} - CONCN_{control}) \times 10/FW \tag{7}$$

where $CONCN_{Fe}$ and $CONCN_{control}$ are the Fe concentration ($\mu g\,mL^{-1}$) of filtrate of sample and control, respectively, and FW is the fresh weight of the young leaves. Multiplication with 10 is applied to account for the volume of hydrochloric acid.

## Available iron in the rhizosphere

10 g of air-dried rhizoplane soil was weighed and added to 20 mL of diethylenetriaminepentaacetic acid (DTPA) extracting agent, containing 0.005 M DTPA, 0.01 M $CaCl_2$ and 0.1 M triethanolamine. Then, the mixture was shaken for 2 h at 180 r.p.m. at 25 °C. Subsequently, the solution was filtered and the iron concentration in the filtrate was measured by ICP-OES. Available iron in the rhizosphere was calculated with the formula (8) below and deionized water was used as a control:

$$\text{Available iron in the rhizosphere}(\mu g\,g^{-1}) = (CONCN_{Fe} - CONCN_{control}) \times 20/weight \tag{8}$$

where $CONCN_{Fe}$ and $CONCN_{control}$ are Fe concentration ($\mu g\,mL^{-1}$) of the soil sample and control, weight is the weight of the soil sample. Multiplication with 20 is applied to account for the volume of DTPA-extracting agents.

## Soil DNA extraction, 16S rRNA amplicon sequencing

16S rRNA amplicon sequencing was performed to examine the impact of intercropping on rhizosphere microbiome compositions. As with any microbiome sequencing study, it is important to consider potential contamination sources. We can identify two such potential sources in our rhizosphere microbiome sample collection scheme, which are (a) incomplete root separation and (b) soil mixing during plant harvest. For (a), it could be that peanut and maize roots cannot fully reliably be distinguished and separated when collecting them. We can refute this possibility because the roots of peanuts and maize can be clearly distinguished based on their color: roots of maize are white, while roots of peanut are dark brown (Supplementary Fig. 25). This distinct difference allowed us the completely and reliably separate the roots of the two species. For (b), there is the possibility that loose soil particles originally associated with the root of one of the species are transferred to the soil collected from the other species during the harvesting process. To prevent this source of contamination, we first vigorously shook the roots of the collected plant to get rid of all loose soil particles that are typically associated with the sandy soil. Subsequently, we only collected the rhizosphere soil that was tightly attached around the peanut or maize roots. These two points ensure to reduce the contamination risk to a minimum and that the convergence we observed stems from the biological interactions between the fine roots of the two plant species. DNA was isolated from 500 mg of soil sample using the FastDNA® Spin Kit for Soil (MP Biomedicals, Solon, USA) according to the manufacturer's instructions. The hypervariable region V3–V4 of the bacterial 16S rRNA gene was amplified with primer 338F (5′-ACTCCTACGGGAGGCAGCAG-3′) and 806R (5′-GGAC-TACHVGGGTWTCTAAT-3′) by an ABI GeneAmp® 9700 PCR thermocycler (Applied Biosystems, Foster City, USA). PCR conditions were as follows: 3 min at 95 °C, then 27 cycles of each 30 s at 95 °C, 30 s at 55 °C and 30 s at 72 °C. After these cycles, there was a 10-min elongation time at 72 °C. Purified amplicons were pooled in equimolar and paired-end sequenced on an Illumina MiSeq PE300 platform (Illumina, San Diego, USA) according to the standard protocols by Majorbio Bio-Pharm Technology Co. Ltd. (Shanghai, China).

The upstream analysis progress of the amplicon sequence was processed using QIIME2 v2021.8[46]. Using the DADA2 workflow in QIIME2, barcodes, PCR primers and low-quality sequences were trimmed, and the remaining reads were subsequently denoised and merged to generate the amplicon sequence variants (ASV) feature table. A taxonomy native Bayes classifier trained on the V3–V4 region (338F and 806R) of the reference 16S sequence (SILVA release 138) was used to assign taxonomic identities to the representative sequences. Only ASV that were at least annotated to the phylum level were retained. Features identified as chloroplast or mitochondrial were removed. Features with a total abundance (summed across all samples) of less than ten or present in less than six samples (10% samples) were filtered. We normalized the sequencing depth to 17,063 counts per sample using scaling with ranked subsampling (SRS)[47], preserving the original community structure by minimizing subsampling errors. Representative sequences were filtered based on the filtered ASV table, then used to generate an approximately-maximum-likelihood phylogenetic tree using FastTree 2.1.8 in QIIME2 with default parameters.

PCoA analysis and Bray-Curtis distance calculation were performed by R 'phyloseq' package (version 1.34.0) based on the relative abundance of ASV. Differential microbiota between groups was identified by linear discriminant analysis (LDA) and LDA effect size (LEfSe) with an LDA threshold of 3 and $p$-value threshold of 0.05. LefSe analysis was performed on the Huttenhower lab Galaxy server (http://huttenhower.sph.harvard.edu/galaxy). The format conversion from ASV table to table for Lefse was performed by R 'amplicon' package (version 1.14.2). The correlation between rhizobacteria genus biomark's relative abundance and soil plant iron nutrition was also based on relative abundance and calculated using non-parametric Spearman's rank correlation (presented as Spearman rho) using R 'phyloseq' package (version 2.2.5).

## Isolation of siderophore-secreting rhizobacteria

We isolated siderophore-secreting-rhizobacteria from the rhizosphere of intercropping peanut to assess the role they play in intercropping peanut iron nutrition improvement. The isolation method was modified according to protocols from the previous literature[48] (Fig. 4a) and details are described below. Briefly, 1 g of the soil sample from intercropping peanut rhizosphere was moderately agitated in 20 mL of sterile 0.9% NaCl solution for 10 min first. Then the microbial suspension (serially diluted) was spread on agar plates with 1% (v/v) chrome azurol S (CAS) indicator. CAS indicator contained 1 mM CAS, 0.1 mM $FeCl_3$; 4 mM hexadecyltrimethylammonium bromide (HDTMA). Other components of the CAS agar medium consisted of (mg $L^{-1}$): sucrose, 2000; casein acid hydrolysate, 3; $CaCl_2$, 222; $MgSO_4$, 241; agar, 1800 and 0.05% (v/v) 0.1 mM phosphate buffer solution (24.27 g $L^{-1}$ $Na_2HPO_4 \cdot 12H_2O$; 5.905 g $L^{-1}$ $NaH_2PO_4 \cdot 2H_2O$, 0.75 g $L^{-1}$ $KH_2PO_4$, 2.5 g $L^{-1}NH_4Cl$, 1.25 g $L^{-1}$ NaCl, pH = 6.8). The pH of the medium was adjusted to 7.0. The ternary complex CAS-Fe(III)-HTDMA forms a blue dye. Siderophores remove Fe(III) from the dye by chelating Fe(III), which induces a color change from blue to orange (in agar plate) or red (in solution)[45]. The plates were incubated at 30 °C for 48 h in the dark. The microbial colonies with or without an orange halo were picked. The isolates were then re-streaked on LB plates for colony purification. The final collection consisted of 324 siderophore-secreting rhizobacteria isolates. All purified isolates were cultured in 1 mL liquid LB medium at 30 °C with shaking (rotary shaker at 180 r.p.m.) before freezing and storing at −80 °C in 15% glycerol. Requests for strains should be addressed to zuoym@cau.edu.cn.

## Measuring the siderophore-secreting capacity of rhizobacteria

To quantify the siderophore-secreting capacity of each rhizobacterium isolate under iron-limited conditions, the rhizobacteria isolates were revived by transferring 50-μL volumes of their freezer stocks into 5 mL liquid LB medium and cultured 18 h at 30 °C with shaking (rotary shaker at 180 r.p.m.). Then, the 50-μL volumes of revived cultures were transferred into 5 mL liquid standard succinic medium (SSM) without

added iron (6 g L$^{-1}$ K$_2$HPO$_4$, 3 g L$^{-1}$ KH$_2$PO$_4$, 1 g L$^{-1}$ (NH$_4$)$_2$SO$_4$, 0.2 g L$^{-1}$ MgSO$_4$·7H$_2$O, 4 g L$^{-1}$ succinic acid, pH = 7.0). After 48-h incubation at 30 °C, the bacterial cultures were diluted to OD$_{600}$ = 0.8. Then, cell-free supernatant from diluted bacterial culture was harvested by centrifugation (12,096 × $g$, 10 min at 4 °C) and filtration (using a 0.22 μm filter). Siderophore concentration in the supernatant was assayed using a modified version of the universal chemical CAS assay. Briefly, CAS assay solution was prepared as follows: a 21.9 mg quantity of HDTMA was dissolved in 25 mL water while constantly stirring over low heat. 1.5 mL of 1 mM FeCl$_3$·6H$_2$O (in 10 mM HCl) was mixed with 7.5 mL of 2 mM CAS. This solution was slowly added to the HDTMA solution while stirring, and the mixture volume was made up to 100 mL with deionized water. Cell-free supernatant or deionized water as a control reference was mixed with an equal volume of CAS assay solution in a 96-well plate and incubated for four h at room temperature (Supplementary Fig. 8). Siderophores removed the Fe(III) from pigmented Fe(III)-CAS-HDTMA complex, inducing a color change from blue to red in the CAS medium solution, lowering the OD$_{630}$ measurements. The OD$_{630}$ of the cell-free supernatants (A) and deionized water controls (Ar) was then measured using a plate reader (SpectraMax M5, Molecular Devices, Sunnyvale, USA). Siderophore-secreting capacity can be quantified using the following formula: Su = (Ar−A) ÷ Ar. This formula quantifies the relative reduction in the OD$_{630}$ measurement induced by iron-chelating compounds that retrieve Fe(III) from the pigmented Fe(III)-CAS-HDTMA complex. The Su-value exhibits exponential instead of linear changes when the siderophore is excessive (Su > 0.73). Thus, the supernatant for the CAS assay reaction solution should be gradient diluted by 1/2 CAS assay solution to ensure Su < 0.73. Desferrioxamine B (DFOB), a purchasable siderophore, was used to establish a calibration curve to estimate the concentration of siderophores in the supernatants of the rhizobacteria. Serial dilutions (1, 3, 6, 12, and 24 μM) of the DFOB standard were mixed with CAS assay solution and incubated for four h at room temperature before OD$_{630}$ determination. The Su-value of 24 μM DFOB was 0.71, and the calibration curve equation was: siderophore equivalent = 32.99 × Su + 27.46 ($R^2$ = 0.999, $p$ < 0.001). Results are expressed as μM equivalents of DFOB. To calculate the iron-binding capacity of bacterial siderophores, one mmol of DFOB is considered to bind one mmol Fe(III).

### Tree construction and phylogenetic analysis of rhizobacterial siderophore-secreting isolates

We sequenced the 16S rRNA genes of the siderophore-secreting rhizobacterial isolates to confirm their genus-group and evolutionary relationships. According to the manufacturer's protocol, the genomic DNA was extracted using Wizard® Genomic DNA Purification Kit (Promega, Charbonnières, France). Purified genomic DNA was quantified by a TBS-380 fluorometer (Turner BioSystems Inc., Sunnyvale, USA). High-quality DNA (OD$_{260/280}$ = 1.8-2.0, >20 μg) was used to do further research. The 16S rRNA gene was amplified using 27 F (5′-AGAGTTTGATCMTGGCTCAG-3′) and 1492 R (5′-TACGGY-TACCTTGTTACGACTT-3′) primers by the polymerase chain reaction (PCR). The PCR reactions (25 μL) contained one μL bacterial DNA, 12.5 μL master mix, one μL each of the forward and reverse primers, and 9.5 μL deionized water. The PCR was run as follows: initial: denaturation at 95 °C for 5 min, 30 cycles of denaturation at 94 °C for 30 s, annealing at 58 °C for 30 s, extension at 72 °C for 1 min 30 s, and the final extension at 72 °C for 10 min. The PCR products were sequenced by Beijing Genomics Institution Co., Ltd (Beijing, China).

The 16S rRNA gene sequences were trimmed at both ends to obtain maximum overlap using the software DNAStar (version 7.1.0, DNASTAR Inc., Madison, USA). The 16S rRNA gene sequences and homologous sequence similarity were identified using the NCBI database (https://www.ncbi.nlm.nih.gov/). We constructed an

approximately-maximum-likelihood phylogenetic tree using Fas-tTree 2.1.8 in QIIME2 with default parameters. Trees were visualized using ggtree v2.4.1 in R v4.2.2.

### Total genomic DNA extraction and sequencing of rhizobacteria isolate

The genome of *Pseudomonas* 1502IPR-01 isolate, which had the highest siderophore-secreting capacity in all rhizobacteria isolates, was sequenced using a combination of PacBio RS II Single Molecule Real Time (SMRT) and Illumina sequencing platforms. The Illumina data were used to evaluate the complexity of the genome.

For Illumina sequencing, at least 1 μg genomic DNA was used in sequencing library construction. DNA samples were sheared into 400–500 bp fragments using a M220 Focused Acoustic Shearer (Covaris, Woburn, USA) following the manufacturer's protocol. Illumina sequencing libraries were prepared from the sheared fragments using the NEXTflex™ Rapid DNA-Seq Kit (Bioo Scientific, Austin, USA). Briefly speaking, 5′ prime ends were first end-repaired and phosphorylated. Next, the 3′ ends were A-tailed and ligated to sequencing adapters. The third step was to enrich the adapters-ligated products using PCR. The prepared libraries were then used for paired-end Illumina sequencing (2 × 150 bp) on an Illumina HiSeq X Ten machine (Illumina, San Diego, USA).

For Pacific Biosciences sequencing, an aliquot of 15 μg DNA was spun in a Covaris g-TUBE (Covaris, Woburn, USA) at 4355 × $g$ for 60 s using an Eppendorf 5424 centrifuge (Eppendorf, NY, USA). DNA fragments were then purified, end-repaired, and ligated with SMRTbell sequencing adapters following the manufacturer's recommendations (Pacific Biosciences, Menlo Park, USA). Following the manufacturer's recommendations, the resulting sequencing library was purified three times using 0.45 × volumes of Agencourt AMPure XP beads (Beckman Coulter Genomics, Danvers, USA). Next, a-10 kb insert library was prepared and sequenced on one SMRT cell using standard methods.

### Genome assembly

The complete genome sequence of *Pseudomonas* sp. 1502IPR-01 was assembled using both the PacBio and Illumina reads. The original image data was transferred into sequence data via base calling, defined as raw data, and saved as a FASTQ file. Those FASTQ files contain read sequences, and quality information is included. A statistic of quality information was applied for quality trimming, by which the low-quality data can be removed to form clean data. The reads were then assembled into a contig using a hierarchical genome assembly process (HGAP) and canu, a software that performs scalable and accurate long-read assembly via adaptive k-mer weighting and repeat separation. The last circular step was checked and finished manually, generating a complete genome with seamless chromosomes and plasmids. Finally, error correction of the PacBio assembly results was performed using the Illumina reads using Pilon. All of the above analyses were performed using I-Sanger Cloud Platform (www.i-sanger.com) from Shanghai Majorbio.

### Phylogenetic tree construction of the 1502IPR-01 and representatives of *Pseudomonas* spp

To confirm the exact evolutionary relationships of 1502IPR-01 and representatives of *Pseudomonas* spp., a multi-locus sequence analysis-based phylogenetic tree was generated using a previously established method[49]. Specifically, TBLASTN+ (v. 2.10.0) was used to identify and extract sequences for 100 housekeeping genes used in Davis et al.[49]. All 100 housekeeping genes of 1502IPR-01 and 23 completely sequenced *Pseudomonas* spp. were aligned using MAFFT (v.7.427). Phylogenetic tree construction was done using IQ-TREE2. Trees were visualized using ggtree v2.4.1 in R v4.2.2. Information of used sequenced *Pseudomonas* spp. strains were listed in Supplementary Table 7.

## Pyoverdine biosynthesis and transport gene cluster analysis

The pyoverdine biosynthesis and transport genes locus in 1502IPR-01 genome were identified using TBLASTN+ (v. 2.10.0) with the protein sequences from *Pseudomonas aeruginosa* PAO1 used as queries against the predicted protein sequences from 1502IPR-01 genome as the databases. Sequence comparisons were conducted using Smith-Waterman algorithm by SnapGene software (version 6.0.2, GSL Biotech LLC, Boston, USA) to obtain DNA sequence identity. Visualization was performed by Adobe Illustrator CC, 2020 (Adobe, San Jose, USA).

## Pyoverdine production and purification

For pyoverdine production, *Pseudomonas* sp. 1502IPR-01 was cultured in an above-mentioned SSM medium. After culturing at 30 °C at 180 r.p.m for 48 h, the culture media (10 L) was isolated from the cell materials by centrifugation at 12,096 × *g* for 15 min and filtration with filter paper. The crude culture extracts were adjusted to pH 5.5 by the addition of HCl and adsorbed by XAD-4, a polymeric adsorbent resin. After removal of supernatant, the resin was then filtered, and the pyoverdine-containing fraction was subsequently eluted with MeOH-$H_2O$ 1:1 (v/v). The MeOH-$H_2O$ 1:1 (v/v) elute was evaporated to dryness in a vacuum. Then the residues were dissolved in 20 mL $H_2O$ and subjected to the Shimadzu semi-preparative HPLC (Shimadzu, Tokyo, Japan) at 20 °C with a flow rate of 3.5 mL min$^{-1}$. The injection volume was 100 μL. Samples were chromatographed using a solvent system of 0.1% trifluoroacetic acid (TFA) in $H_2O$ (solvent A) and MeCN-MeOH 4:1 (v/v, solvent B). A linear binary gradient-0–15 min 11.0–12.5% B, 15–20 min 12.5–13.0% B, 20–25 min 13.0–100.0% B-was used. The eluate was monitored at 380 nm. The faction containing pyoverdine was collected at 13–14.5 min and then adjusted to pH 7.0 with 0.1% ammonium hydroxide before evaporation. The salt in the fraction generated during the neutralization of TFA was removed by a Spectra/Por Dialysis membrane with a molecular weight cut-off (MWCO): 500–1000 Da (Spectrum Lab., Rancho Dominguez, USA). Then, the purity > 99.5% of isolated pyoverdine was confirmed by HPLC.

## Structural identification of pyoverdine

**Mass spectrometry**. LC-HRMS spectra were obtained on an Agilent 6530 Q-TOF mass spectrometer coupled to an Agilent 1260 HPLC (Agilent Technologies GmbH, Waldbronn, Germany); Thermo Orbitrap MS (Orbitrap Fusion Lumos, Thermo Fisher Scientific, San Jose, USA); ESI-QQQ-MSMS (Xevo TQ-S micro, Waters, Milford, USA).

NMR data were obtained using (operating at 500.13 MHz for $^1$H-NMR and 125.75 MHz for $^{13}$C-NMR) (Bruker Biospin GmbH, Karlsruhe, Germany). All spectra were measured at 303 K. The residual solvent signals were used for referencing spectra in the $^1$H and $^{13}$C dimensions.

**UV/Vis**. The UV spectra were recorded by a SPD-M20A Shimadzu photodiode array detector (PDA, detection 190–800 nm) (Shimadzu, Kyoto, Japan) connected to a Shimadzu HPLC system (LC-20AR, Shimadzu, Japan) during semi-preparative HPLC separations in MeCN-MeOH-$H_2O$ containing 0.1% TFA in $H_2O$.

**Semi-preparative HPLC**. Semi-preparative HPLC separations were conducted on a Shimadzu HPLC system (LC-20AR, Shimadzu, Japan) with an Atlantis Prep OBD T3 column (5 μm, 150 × 19 mm, Waters, USA).

**Chemicals**. Ultrapure water was acquired from Watsons Ltd. (Hong Kong, China). Acetonitrile (HPLC grade) and TFA (HPLC grade) used for UHPLC/Q-TOF-MS were purchased from Merck (Darmstadt, Germany) and Roe Scientific Inc. (Newark, USA), respectively. Sodium carbonate (HPLC grade) was purchased from Sigma-Aldrich (St. Louis, USA). The dialysis membrane was purchased from Spectrum Laboratories, Inc. (Rancho Dominquez, USA).

Pyoverdine (Supplementary Fig. 11a), obtained as a yellowish-green crystalline powder, showed the typical UV/Vis spectra of pyoverdines: 387 nm at pH 7.0 and a split band of 365 and 407 nm at pH 3.0 and 10.0. The molecular formula of $C_{57}H_{84}N_{16}O_{25}$ was determined by the pseudo-molecular ion peak at *m/z* 1393.5876 [M + H]$^+$ (calculated for $C_{57}H_{85}N_{16}O_{25}{}^+$, 1393.5872) in positive HR-ESI-MS (Supplementary Fig. 11b). Total hydrolysis of pyoverdine indicated a peptide chain containing the three types of amino acids: serine (Ser), lysine (Lys) and 5-N-formyl-5-N-hydroxyornithine (FoOHOrn) (Supplementary Fig. 12).

It has been well documented that mass spectrometry is a powerful method for structural elucidation of pyoverdines[50]. Herein, we obtained the following characteristic fragment ions of pyoverdine by using an orbitrap higher-energy collisional dissociation (HCD) approach with varying collision energies. An ion at *m/z* 204.08, the most characteristic ion for the pyoverdine chromophore (**A1'-2**, Supplementary Fig. 11c), was clearly recognized in the orbitrap MS (Supplementary Fig. 11d). Nearby, another abundant ion at *m/z* 270.09, corresponding to the chromophore connected with a carboxylic amide side chain (**A1'-1**) (Supplementary Fig. 11c, d), could be explained as the fragment ion of the opening of the tetrahydropyrimidine ring of the chromophore by *retro*-Diels-Alder (RDA) reaction. Then, the side chain ketoglutaric acid (kgl) of pyoverdine was evidenced by the loss of $[CO_2 + H_2O]$ from the A1 (*m/z* 445.14) and B1 (*m/z* 473.13) ions yielding the two corresponding **A1'** (*m/z* 383.14) and **B1'** (*m/z* 411.13) ions (Supplementary Fig. 11e)[50]. Hence, the structure-fragmentation pathways from **B1** to **A1'-2** were proposed as shown in Supplementary Fig. 11c. Furthermore, the series of B-ions from **B8** (*m/z* 1265.49) to **B1** (*m/z* 473.13) allowed to determine of the amino acid sequence of the peptide chain as Ser–Ser–FoOHOrn–Ser–Ser–(Lys–FoOHOrn–Lys–Ser) (Supplementary Figs. 12–14). The series of B'-ions (lossing of $H_2O + CO_2$ from the kgl chain of the B-ions, Supplementary Figs. 12–14) form **B8'** (*m/z* 1203.49) to **B1'** (*m/z* 411.13) corroborated the peptide sequence. Taken together, the primary structure of pyoverdine was deduced as kgl-chr-Ser–Ser–FoOHOrn–Ser–Ser–(Lys–FoOHOrn–Lys–Ser), which is identical to the pyoverdine from *Pseudomona*s spp. CFML 95-275[51].

The structure of pyoverdine deduced from MS data was further confirmed by the NMR data. On the basis of a combination of 1D ($^1$H- and $^{13}$C-NMR) and 2D (COSY, NOESY, HSQC, and HMBC) NMR experiments (Supplementary Figs. 15–20), the $^1$H- and $^{13}$C-data of pyoverdine were assigned in Supplementary Tables 4 and 5. In addition, the key COSY and HMBC correlations are shown in Supplementary Fig. 21. The results showed that $^1$H-NMR data of pyoverdine was almost identical to that of the reported pyoverdine[51]. While the $^{13}$C-NMR chemical shift values of pyoverdine shifted 1–2 ppm, which could be caused by the different acquisition temperatures or machines. The pyoverdine molecule consists of three parts: (i) a dihydroquinoline-type chromophore (chr): (1S)-5-amino-2,3-dihydro-8,9-dihydroxy-1H-pyrimido-[1,2-a]quinolone-1-carboxylic acid, (ii) a peptide comprising 9 amino acids: Ser-Ser-FoOHOrn-Ser-Ser-(Lys-FoOHOrn-Lys-Ser) (Ser: serine; FoOHOrn: 5-N-formyl-5-N-hydroxyornithine; Lys: lysine), and (iii) a side chain (α-ketoglutaric acid, kgl) bound to the nitrogen atom at position C-3 of the chromophore.

## The capacity of pyoverdine of *Pseudomonas* sp. 1502IPR-01 to solubilize iron from Fe(OH)$_3$

Fe(OH)$_3$ is the main insoluble form of iron in calcareous soil. Therefore, we determined the capacity of the pyoverdine of *Pseudomonas* sp. 1502IPR-01 to solubilize iron from insoluble Fe(OH)$_3$ stocks according to previous paper[52]. For this purpose, 0, 0.025, 0.2, 0.5, 1 μmol equivalent of purified pyoverdine (calculated based on the DFOB calibration curve) was added into 2 mL 0.5 M sodium acetate solution. Then, all solutions were added to 0.5 mL 5 mM Fe(OH)$_3$ (Fe(OH)$_3$ solution was prepared by neutralizing FeCl$_3$ with NaOH to pH 6.5–7.0) suspensions to a total volume of 5 mL. The above solutions were

incubated at 55 °C for five hours, and the supernatant was collected after centrifugation at $1000 \times g$ at 25 °C for 10 min. Finally, the iron content in the supernatant was determined by ICP-OES using a 7300DV system (Perkin Elmer, Waltham, USA).

### Functional verification of *Pseudomonas* sp. 1502IPR-01 and its siderophore

We identified the effect of *Pseudomonas* sp. 1502IPR-01 and its siderophore on peanut iron nutrition improvement in both greenhouse and field conditions. The normal and sterilized iron-deficient calcareous sandy soil was prepared as described above for pot experiment. Peanut (*Arachis hypogaea* L. cv. Luhua14) and maize (*Zea mays* L. cv. Zhengdan958) were used. Four treatment groups were set up: (1) monocropping peanut grown in normal soil; (2) monocropping peanut grown in sterilized soil; (3) peanut/maize intercropping grown in normal soil; (4) peanut/maize intercropping grown in sterilized soil. Six monocropping peanut plants were grown in a pot for the monocropping peanut treatment, and three peanut plants with three maize plants were grown in a pot for the intercropping treatment. Each pot contained 8 kg of soil. The rest of the greenhouse culture conditions remained identical as described above. At anthesis (50 dps), pod bearing (64 dps), and fruit-swelling stage (75 dps), 600 mL of $10^8$ CFU mL$^{-1}$ 1502IPR-01 strain solution ($1 \times 10^{10}$ CFU plant$^{-1}$) or 600 mL of 20 μM purified pyoverdine (2 μmol plant$^{-1}$) solution were irrigated into the rhizosphere as 1502IPR-01 and pyoverdine treatment. The 600 mL of water was irrigated into the rhizosphere as control. It is worth noting that we applied 20 μM pyoverdine solution to treat the rhizosphere of peanut according to previous literature[21] and our own research experience[6]. The peanut plants in the same pot were mixed as a replicate. Each treatment had three or four replicates. At 84 dps, SPAD values and active iron concentration in young leaves were measured. Rhizosphere soil was collected to measure the available iron concentration.

The field experiment was conducted in Xichang village (116°10′ E, 39°39′ N), Fangshan district of Beijing, and Ligu village (115°9′ E, 36°7′ N) in Puyang city, Henan province, China in 2020. In Beijing, the peanut cultivar was Luhua14. The average temperature from May to September was 25.6 °C, and the annual precipitation was 603 mm. The type of soil in Beijing is calcic cambisol. The soil property was pH 7.9, total N 0.038%, available P (Olsen-P) 5.12 mg·kg$^{-1}$, available K (NH$_4$OAc-K) 60.32 mg·kg$^{-1}$, available Fe 3.35 mg·kg$^{-1}$. In Henan, the peanut cultivar was Yuhua37. The average temperature from May to September was 25.4 °C, and the annual precipitation was 604.1 mm. The type of soil in Henan is fluvo-aquic soil. The soil property was pH 7.7, total N 0.041%, available P (Olsen-P) 7.42 mg·kg$^{-1}$, available K (NH$_4$OAc-K) 70.2 mg·kg$^{-1}$, available Fe 3.74 mg·kg$^{-1}$. 900 kg hm$^{-2}$ compound fertilizer (N-P-K: 19-19-19) was applied before sowing, and field management was according to local practices. Four treatments were set up: (1) 150 mL water was irrigated in the rhizosphere, and 16 mL water was sprayed on young leaves for one peanut plant as a control treatment; (2) 150 mL $10^8$ CFU mL$^{-1}$ 1502IPR-01 strain solution was irrigated in rhizosphere, and 16 mL water was sprayed on young leaves for one peanut plant as 1502IPR-01 treatment; (3) 150 mL 48 μM purified pyoverdine (7.2 μmol plant$^{-1}$) was irrigated into rhizosphere, and 16 mL water was sprayed on young leaves for one peanut plant as pyoverdine treatment; (4) 150 mL water was irrigated in the rhizosphere and 16 mL 48 μM EDTA-FeNa solution was sprayed on young leaves for one peanut plant as EDTA-Fe treatment. All spraying solutions contained 0.1% Triton X-100 as surfactant. Each treatment consisted of six field plot replications in Beijing and eight field plot replications in Puyang. Each field plot replication consisted of 16 holes, and two peanut plants were grown in one hole. The plant spacing was 30 cm, and row spacing was 60 cm. At anthesis (53 dps in Beijing and Puyang), pod bearing (65 dps in Beijing and 68 dps in Puyang), and fruit-swelling (79 dps in

Beijing and 84 dps in Puyang) stage, the respective treatments were repeated. Two weeks after the third treatment, SPAD values and active iron concentration in young leaves were measured. Rhizosphere soil was collected to measure the available iron concentration. Then, peanut plants were harvested to measure yield.

### Pot experiment with pyoverdine null mutant

To further strengthen the point that pyoverdine is produced in soil and involved in peanut iron nutrition, we conducted additional experiments with the laboratory strain *P. aeruginosa* PAO1 and its defined isogenic pyoverdine null mutant PAO1 Δ*pvdDpchEF*. PAO1 Δ*pvdDpchEF* is a double knockout mutant of the primary siderophore pyoverdine (through the deletion of *pvdD*) and the secondary siderophore pyochelin (through the deletion of *pchE* and *pchF*) in *P. aeruginosa* PAO1. We used this double knockout mutant because pyochelin is upregulated when pyoverdine is knocked out and can compensate for the effect of pyoverdine. To assess the role of pyoverdine, we thus needed to use the double knockout mutant. The normal iron-deficient calcareous sandy soil was prepared as described above for the pot experiment with each pot containing 8 kg of soil. Six monocropping peanut plants (*Arachis hypogaea* L. cv. Luhua14) were grown in a pot for the monocropping peanut treatment. At anthesis (50 dps), pod bearing (64 dps), and fruit-swelling stage (75 dps), 600 mL of $10^8$ CFU mL$^{-1}$ ($1 \times 10^{10}$ CFU plant$^{-1}$), solutions of *Pseudomonas* sp. 1502IPR-01, *P. aeruginosa* PAO1 wildtype, and the siderophore-deficient mutant PAO1 Δ*pvdDpchEF* were irrigated into the rhizoplane. 600 mL of water was irrigated into the rhizosphere as control. The rest of the greenhouse culture conditions remained identical to the ones described above. The peanut plants growing in the same pot were mixed and treated as one replicate. Each treatment had three or four replicates. At 84 dps, SPAD values and active iron concentration in young leaves were measured. Rhizosphere soil was collected to measure the available iron concentration.

### Data analysis

Data sets were tested for normal distribution with the Shapiro-Wilk test and for heterogeneity of variance using Levene's test. For data sets that met the conditions of normal distribution and homogeneity of variance, parametric Student's $t$-test and ANOVA with LSD as post-hoc test were used. For data sets that did not meet these conditions, the BoxCox algorithm was used for data transformation to meet the assumptions of parametric tests. If these conditions were still not met after transformation, non-parametric Wilcoxon tests, Kruskal–Wallis tests with Dunnett T3 test (for one-way designs) or Scheirer-Ray-Hare test (for two-way designs) were used. The BH algorithm was used to correct $p$-values in multiple comparisons. All the above tests, as well as PERMANOVA and correlation analyses, were analyzed in R (version 4.2.2; https://www.r-project.org) using the 'stats' (version 4.2.2), 'car' (version 3.1-1), 'forecast' (version 8.2.0), 'ggpubr' (version 0.4.0), 'stats' (version 4.2.2), 'agricolae' (version 1.3-5), 'ggpubr' (version 0.4.0), 'stats' (version 4.2.2), 'PMCMRplus' (version 1.9.6), 'rcompanion' (version 2.4.21), 'vegan' (version 2.5-7), and 'psych' (version 2.2.5) packages. For data sets that did not meet normal distribution, correlations were calculated using non-parametric Spearman's rank correlation (presented as Spearman rho). For data sets that met normal distribution, correlations were calculated using parametric Pearson correlation (presented as Pearson's r). The heatmap was generated using 'ComplexHeatmap' (version 2.15.1) package in R (version 4.2.2) based on the relative abundance of the genus. Then, plots including conceptual diagrams were assembled and aesthetically perfected by Adobe Illustrator CC, 2020 (Adobe, San Jose, USA). All tests were two-sided. The China administrative border map used in the figures was downloaded from a public platform: resource and environment center data cloud platform (http://www.resdc.cn/).

**Reporting summary**

Further information on research design is available in the Nature Portfolio Reporting Summary linked to this article.

## Data availability

Raw data of 16S rRNA amplicon sequencing used in this study are deposited in the NCBI database under BioProject PRJNA788265. 16S rRNA gene sequences of 46 isolated rhizobacteria are deposited on GenBank with accession OL824679 - OL824724. The *Pseudomonas* sp. 1502IPR-01 reference strain genome sequence, determined for this study, is deposited at the NCBI database under BioProject PRJNA788188. The raw data generated in this study are available in the Figshare repository under accession code (https://doi.org/10.6084/m9.figshare. 24648306)[53] and also are provided in the Source Data file/Supplementary Data with this paper. Source data are provided with this paper.

## Code availability

The analysis code that supports the findings of this study is available on GitHub (https://github.com/wnq13579/Peanut-maize-intercropping-microbiome) and deposited in Zenodo with https://doi.org/10.5281/zenodo.10212602[54] (https://zenodo.org/records/10212602).

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

## Acknowledgements
This research was financially supported by the National Natural Science Foundation of China (grant no. 32372810 to Y.-M. Z., 32102469 to N.-Q. W., and 42325704 to Z. W.), the National Key Research and Development Program of China (grant no. 2022YFD1901500/2022YFD1901501, 2023YFD1700203 to Y.-M. Z. and 2021YFD1900100 to Z. W.), and the Swiss National Science Foundation SNSF (grant no. 31003A_182499 to R. K.). We thank Dr. Prof. Yang Bai from the Chinese Academy of Sciences, Dr. Prof. Chuihua Kong, and Dr. Prof. Ling Xu from China Agricultural University for valuable discussions and suggestions.

## Author contributions
Y.-M. Z., Z. W., N.-Q. W. and T.-Q. W. designed the research. N.-Q. W. and T.-Q. W. performed and analyzed most of the experiments in the laboratory and field, and Q.-F. L. K.-G. W., Z.-C. D., Z.-G. C., W.-Q., J.-D., L.-N., and J.-Y. C. performed and analyzed most of the experiments in the greenhouse. Y. C. and M. W. provided lab conditions and intellectual input for identifying the siderophore structure and wrote the section "Structural identification of pyoverdine" in methods. R. K. and Z. W. provided intellectual input, and guidance to microbiome data analysis and helped to interpret data. M. W. provided some intellectual input for this manuscript, and guidance to microbiome data analysis and visualization. N.-Q. W., T.-Q. W., F.-S. Z., R. K., Z. W., and Y.-M. Z. wrote and revised the manuscript. All of the authors discussed the results and commented on the manuscript. Y.-M. Z. supervised the study.

## Competing interests
The authors declare no competing interests.
