## [Peer Review File · Nature Communications]

Reviewers' Comments:

Reviewer #1:

General comments:

The authors showed in this study that intercropping peanuts and maize has positive effect on iron acquisition in plants, particularly in peanuts. By carrying out experiments in greenhouse with mixed crops (peanuts and maize) and monoculture (peanuts or maize) with gamma irradiated soil or not, they showed that improved iron nutrition is facilitated by *Pseudomonas*. The authors have done extensive experiments, and *Pseudomonas* strain isolation and characterization (genome sequencing, siderophore production, siderophore types, etc...). They concluded that their *Pseudomonas* 1502IPR-01 strain secretes pyoverdine, what becomes the preferred route of iron acquisition, benefiting both peanut and maize plants. The role of *Pseudomonas* as PGPB is well known; besides traits of siderophore production, they are also capable of solubilize phosphate, produce hormones, secondary metabolites, etc... There are many literature available including on intercropping systems with grass and leguminous crops.

Specific comments:

Results:

L135-144: This network analysis is just a correlation between abundances and do not reflect any ecological relationship between microbes or plant-microbe-plant as mentioned in the abstract. In addition, it also does not exclude the relationship between microbes because of a third factor. Thus, I suggest to delete this result from the manuscript.

L165-170. The authors found their strain *Pseudomonas* 1502IPR-01 similar to *P. extremorientalis* based on complete genome sequences. Later the authors named their *Pseudomonas* 1502IPR-01 strain as *P. extremorientalis* 1502IPR-01. This is not correct giving a taxonomy based on this comparison. The authors should be careful and mention that their *Pseudomonas* 1502IPR-01 is similar to *P. extremorientalis* throughout the paper and not call it *P. extremorientalis* 1502IPR-01.

L197-200: What about competition between microbes and maize for siderophore?

Material and Methods:

Soil chemical factors are not complete in Supplementary Table 8. Total N is not enough. Nitrogen forms like NH_4 , NO_3 should be provided. They play important role in microbes, and microbes interaction with plants. They might have indirect effect on the microbes producing siderophores.

In addition, organic carbon should be provided.

L 477-479: give the PCR conditions or the reference

L767-769: For PCoA and Bray-Curtis, what type of data were used? Relative abundance or normalized data? How the data was normalized? Add information.

Reviewer #2:

Remarks to the Author:

This study is probably the most bottom-up approach to elucidate the mechanisms involved in the beneficial actions at work in the interactions between crops in intercropping. The manuscript details first how the maize-peanut intercropping (IM) improves the iron nutrition of peanut (a strategy I plant). The involvement of microbiome in this effect is validated and the authors show that intercropping causes shifts in the microbiome composition. They further analyzed the microbiome composition using 16S sequencing and identified several beneficial genera and *Pseudomonas* in particular. One *Pseudomonas* species was identified after a phylogenetic analysis as being *P. extremeorientalis*, which shows high siderophore production. Whole genome sequencing allowed the identification of genes involved in pyoverdine siderophore biosynthesis and uptake. Further, the pyoverdine from *P. extremeorientalis* was purified and its structure determined. Pot and field experiments revealed that *P. extremeorientalis* and its purified pyoverdine could improve the iron nutrition.

The manuscript is very well written and the figures are very instructive. The discussion is very interesting and suggests the involvement of other bacterial genera in the positive interaction that deserve further investigations. The data presented in supplementary figures complete nicely those presented in the main manuscript.

I have however few remarks:

- The use of a mutant of *P. aeruginosa* unable to produce any siderophore confirms the importance of pyoverdines whatever the differences in peptide chain in the beneficial effects observed, but a siderophore negative mutant of *P. extremeorientalis* should have been constructed as well if possible.
- In the same line: some genes involved in the synthesis of the pyoverdine peptide chain (NRPS) are presented, but it would be nice to present the entire locus, including *pvdL* (chromophore biosynthesis), *pvdS* (ECF sigma), *fpvA* (receptor) and other genes involved in uptake and synthesis as a supplementary figure.
- *Pseudomonas* often produce a second siderophore next to pyoverdine. Is it the case for *P. extremeorientalis*? This is not mentioned in the manuscript. If it is the case it could justify why the construction of a siderophore negative mutant is difficult.
- Likewise, rhizosphere bacteria, especially pseudomonads, also produce antibiotic compounds able to tackle phytopathogens. Were any found in the genome?

Point-by-point responses to the Reviewers' Comments:

Reviewer #1:

General comments

The authors showed in this study that intercropping peanuts and maize has positive effect on iron acquisition in plants, particularly in peanuts. By carrying out experiments in greenhouse with mixed crops (peanuts and maize) and monoculture (peanuts or maize) with gamma irradiated soil or not, they showed that improved iron nutrition is facilitated by *Pseudomonas*. The authors have done extensive experiments, and *Pseudomonas* strain isolation and characterization (genome sequencing, siderophore production, siderophore types, et). They concluded that their *Pseudomonas* 1502IPR-01 strain secrete pyoverdine, what becomes the preferred route of iron acquisition, benefiting both peanut and maize plants. The role of *Pseudomonas* as PGPB is well known; besides traits of siderophore production, they are also capable of solubilize phosphate, produce hormones, secondary metabolites, etc. There are many literature available including on intercropping systems with grass and leguminous crops.

Specific comments:

Results:

Comment 1: L135-144: This network analysis is just a correlation between abundances and do not reflect any ecological relationship between microbes or plant-microbe-plant as mentioned in the abstract. In addition, it also does not exclude the relationship between microbes because of a third factor. Thus, I suggest to delete this result from the manuscript.

Response 1: Thank you for raising this point. We agree that network analysis is not so important to figure out the key microbiome members involved in iron nutrition improvement by intercropping. In this context, the results of the LefSe analysis, the relative abundance of biomarkers in the different rhizosphere and their correlation with soil-plant iron nutrition are much more relevant. We have deleted the network analysis from original Fig. 2c,d, Supplementary Fig. 8, Supplementary Table 2 and related figure legends, results, discussion, methods and references.

Comment 2: L165-170. The authors found their strain *Pseudomonas* 1502IPR-01 similar to *P. extremorientalis* based on complete genome sequences. Later the authors named their *Pseudomonas* 1502IPR-01 strain as *P. extremorientalis* 1502IPR-01. This is not correct giving a taxonomy based on this comparison. The authors should be careful and mention that their *Pseudomonas* 1502IPR-01 is similar to *P. extremorientalis* throughout the paper and not call it *P. extremorientalis* 1502IPR-01.

Response 2: We agree with this suggestion. Now we refer to this strain as *Pseudomonas* sp. 1502IPR-01 throughout the manuscript.

Comment 3: L197-200: What about competition between microbes and maize for siderophore?

Response 3: Thank you for raising this important point. We agree that plants and microbes may compete for iron. Clearly, bacteria like *Pseudomonas* spp. primarily secrete pyoverdines to obtain iron for themselves and not to feed plants. However,

siderophores like pyoverdines are highly diffusible such that there is always a certain loss of molecules (i.e., many iron-loaded molecules will not find their way back to bacterial cells), which then become available for plants (Kümmerli, 2023). Moreover, plants secrete nutrient-rich exudates that attract bacteria, potentially fostering a mutually beneficial nutrient vs. siderophore exchange (Kümmerli, 2023; Gu et al., 2020; Harbort et al., 2020; Sasse et al., 2018; Shirley et al., 2011). For these reasons, we believe that there is relatively little competition for iron between plants and bacteria.

Gu, S. et al. (2020). Competition for iron drives phytopathogen control by natural rhizosphere microbiomes. *Nature Microbiology*. **5**: 1002–1010.

Harbort, C.J., Hashimoto, M., Inoue, H., Niu, Y., Guan, R., Rombola, A.D., Kopriva, S., Voges, M., Sattely, E.S., Garrido-Oter, R., and Schulze-Lefert, P. (2020). Root-secreted coumarins and the microbiota interact to improve iron nutrition in *Arabidopsis*. *Cell Host Microbe* **28**: 825–837.

Kümmerli, R. (2023). Iron acquisition strategies in *pseudomonads*: mechanisms, ecology, and evolution. *Biometals* **36**: 777–797.

Sasse, J., Martinoia, E., and Northen, T. (2018). Feed your friends: Do plant exudates shape the root microbiome? *Trends in Plant Science* **23**: 25–41.

Shirley, M., Avoscan, L., Bernaud, E., Vansuyt, G., and Lemanceau, P. (2011). Comparison of iron acquisition from Fe-pyoverdine by strategy I and strategy II plants. *Botany* **89**: 731–735.

Material and Methods:

Comment 4: Soil chemical factors are not complete in Supplementary Table 8. Total N is not enough. Nitrogen forms like NH_4 , NO_3 should be provided. They play important role in microbes, and microbe interaction with plants. They might have indirect effect on the microbes producing siderophores. In addition, organic carbon should be provided.

Response 4: As requested, we have added the data on NH_4^+ -N, NO_3^- -N and organic carbon contents in Supplementary Table 7.

Comment 5: L 477-479: give the PCR conditions or the reference

Response 5: Thank you for your suggestion. We have added the PCR conditions in lines 496-500 of the methods section in manuscript file with the tracked changes.

Comment 6: L767-769: For PCoA and Bray-Curtis, what type of data were used? Relative abundance or normalized data? How the data was normalized? Add information.

Response 6: For PCoA and Bray-Curtis analysis, relative abundance was used without normalization. We have added the information in the legend of Fig. 2 in lines 1000-1010 in manuscript file with the tracked changes

Reviewer #2 (Remarks to the Author):

This study is probably the most bottom-up approach to elucidate the mechanisms involved in the beneficial actions at work in the interactions between crops in intercropping. The manuscript details first how the maize-peanut intercropping (IM) improves the iron nutrition of peanut (a strategy I plant). The involvement of microbiome in this effect is validated and the authors show that intercropping causes shifts in the microbiome composition. They further analyzed the microbiome composition using 16S sequencing and identified several beneficial genera and *Pseudomonas* in particular. One *Pseudomonas* species was identified after a phylogenetic analysis as being *P. extremeorientalis*, which shows high siderophore production. Whole genome sequencing allowed the identification of genes involved in pyoverdine siderophore biosynthesis and uptake. Further, the pyoverdine from *P. extremeorientalis* was purified and its structure determined. Pot and field experiments revealed that *P. extremeorientalis* and its purified pyoverdine could improve the iron nutrition.

The manuscript is very well written and the figures are very instructive. The discussion is very interesting and suggests the involvement of other bacterial genera in the positive interaction that deserve further investigations. The data presented in supplementary figures complete nicely those presented in the main manuscript.

I have however few remarks:

Comment 7: The use of a mutant of *P. aeruginosa* unable to produce any siderophore confirms the importance of pyoverdines whatever the differences in peptide chain in the beneficial effects observed, but a siderophore negative mutant of *P. extremeorientalis* should have been constructed as well if possible.

Response 7: We agree that experiments with a pyoverdine null mutant in the *Pseudomonas* sp. 1502IPR-01 background would be the most appropriate way to provide direct genetic evidence for the effect of pyoverdine. That is the reason why we started to construct such a pyoverdine mutant back in 2020. Unfortunately, we were not successful, and it turned out to be impossible (for us) to genetically modify this natural isolate. Therefore, we had no other choice than to use the PAO1 pyoverdine null mutant as an alternative. *P. aeruginosa* isolates are also frequently found in soil and freshwater habitats, and this species has been used as a valid control in other studies on plant iron nutrition improvement (Hernandez-Calderon et al., 2018). It is a generalist species, thriving in many habitats and that is the reason why we believe that a *P. aeruginosa* pyoverdine null mutant is a valid alternative to provide the first genetic evidence that pyoverdine is involved in iron nutrition improvement.

Hernandez-Calderon, E., Aviles-Garcia, M.E., Castulo-Rubio, D.Y., Macias-Rodriguez, L., Ramirez, V.M., Santoyo, G., Lopez-Bucio, J., and Valencia-Cantero, E. (2018). Volatile compounds from beneficial or pathogenic bacteria differentially regulate root exudation, transcription of iron transporters, and defense signaling pathways in *Sorghum bicolor*. *Plant molecular biology* **96**: 291–304.

Comment 8: In the same line: some genes involved in the synthesis of the pyoverdine peptide chain (NRPS) are presented, but it would be nice to present the entire locus, including pvdL (chromophore biosynthesis), pvdS (ECF sigma), fpvA (receptor) and other genes involved in uptake and synthesis as a supplementary figure.

Response 8: Thank you for your suggestion. We now present the entire locus including pvdL (chromophore biosynthesis), pvdS (ECF sigma), fpvA (receptor) and other genes involved in iron uptake and synthesis. Some genes found in PAO1 have no homologues in *Pseudomonas* sp. 1502IPR-01 and are therefore not shown in the figure. We think that this result is very important for the main message of the paper and that is why we included the new scheme in Fig. 3e and not in a supplementary figure. We have further updated the Supplementary Table 3 and implemented small edits to improve clarity.

Comment 9: *Pseudomonas* often produce a second siderophore next to pyoverdine. Is it the case for *P. extremeorientalis*? This is not mentioned in the manuscript. If it is the case it could justify why the construction of a siderophore negative mutant is difficult.

Response 9: Thank you for raising this important issue. It is correct that certain *Pseudomonas* strains can produce a secondary siderophore, but it is also common that strains only produce pyoverdine. Pyochelin, enantio-pyochelin, quinolobactin, thio-quinolobactin, achromobactin, PDTC (pyridine-2,6-bis(thiocarboxylic acid)), yersiniabactin and pseudomonine are such secondary siderophores (Kümmerli, 2023). That is the reason why we blasted the homologous genes biosynthesizing those siderophores against the genome of *Pseudomonas* 1502IP1-01. There were no hits (Table R1). Moreover, there is no report in the literature about the ability of *P. extremorientalis* to produce a secondary siderophore. Finally, our own data show that pyoverdine explains 88.81-91.96% of the iron-binding capacity of this species (Supplementary Table 4). These pieces of evidence together suggest that *Pseudomonas* 1502IP1-01 has no secondary siderophore and if it has one it is not particularly relevant for iron scavenging.

Table R1 *Pseudomonas* sp. 1502IPR-01 do not secrete secondary siderophores according to genomic search

Siderophore types	Biosynthetic genes name	Gene locus tag	Description	Search results
pyochelin/ enantio-pyochelin	pchG	PA4224	dihydroaeruginic acid synthetase	No hit
	pchF	PA4225	pyochelin synthetase	No hit
	pchE	PA4226	dihydroaeruginic acid synthetase	No hit
quinolobactin/ thio- quinolobactin	QbsN	AAQ21378	MFS permease	No hit
	QbsM	AAL65278	MFS transporter	No hit
	QbsL	AAL65279	AMP-ligase/C-methyl transferase	No hit
	QbsK	AAL65280	oxidoreductase	No hit
achromobactin	AcsD	EPF66772	Achromobactin biosynthesis protein AcsD	No hit
	AcsE	EPF66773	Achromobactin biosynthesis protein AcsE	No hit
	AcsA	EPF66777	Achromobactin biosynthesis protein AcsD	No hit
PDTC (pyridine- 2,6- bis(thiocarboxylic acid))	OrfI	AAQ01712	oxidoreductase	No hit
	OrfJ	AAQ01713	AMP ligase	No hit
yersiniabactin	YbtE	WP_011168856	yersiniabactin biosynthesis salicyl-AMP ligase	No hit
	Irp5	EFW84211	yersiniabactin synthetase, salicylate ligase component	No hit
	Irp4	EFW84210	yersiniabactin synthetase, thioesterase component	No hit
	Irp3	EFW84209	yersiniabactin synthetase, thiazolanyl reductase component	No hit
	EFW84207.1	EFW84207	yersiniabactin non-ribosomal peptide synthetase	No hit
pseudomonine	PmsD	PMSD_PSEE4	non-ribosomal peptide synthase	No hit
	PmsE	PMSE_PSEE4	non-ribosomal peptide synthase	No hit

Kümmerli, R. (2023). Iron acquisition strategies in *pseudomonads*: mechanisms, ecology, and evolution. *Biometals* **36**: 777–797.

Comment 10: Likewise, rhizosphere bacteria, especially pseudomonads, also produce antibiotic compounds able to tackle phytopathogens. Were any found in the genome?

Response 10: Thank you for your suggestion. We used antiSMASH 7.0.1 to predict the secondary metabolites of *Pseudomonas* 1502IPR-01. We found that *Pseudomonas* 1502IPR-01 genome has a gene cluster sharing 68% similarity with the viscosin biosynthetic gene cluster from *Pseudomonas fluorescens* SBW25 (Fig. R1), suggesting that *Pseudomonas* 1502IPR-01 may produce viscosin, a lipopeptide biosurfactant.

Many studies have reported that lipopeptides have antibacterial and antifungal properties. Viscosin and viscosin-like lipopeptides inhibit phytopathogens, like *Rhizoctonia solani* AG2-2(Oni et al., 2020), *Pythium myriotylum* CMR1(Oni et al., 2020) but also human pathogens like Methicillin-resistant *Staphylococcus aureus* (Syaban et al., 2021). While the presence of a viscosin synthesis cluster is interesting, we prefer to not report it in the main manuscript. This is because our paper is already quite dense and viscosin is not the focus of our study.

Fig. R1 *Pseudomonas* 1502IPR-01 may produce viscosin according to genome information. **a** organization of gene cluster in location of 2,588,401-2,695,974 nt in *Pseudomonas* 1502IPR-01 genome. **b** Blast result implies this gene cluster shares 68% similarity with the viscosin biosynthetic gene cluster.

Oni, F.E., Geudens, N., Adiobo, A., Omoboye, O.O., Enow, E.A., Onyeka, J.T., Salami, A.E., De Mot, R., Martins, J.C., and Höfte, M. (2020). Biosynthesis and antimicrobial activity of Pseudodesmin and Viscosinamide cyclic lipopeptides produced by *Pseudomonads* associated with the *Cocoyam* rhizosphere. *Microorganisms* **8**: 1079.

Syaban, M.F.R., Erwan, N.E., Raihan Syamsuddin, M.R., Zahra, F.A., and Sabila, F.L. (2021). Molecular docking approach of Viscosin as antibacterial for methicillin-resistant *Staphylococcus Aureus* Via β -Lactamase inhibitor mechanism. *Clinical and Research Journal in Internal Medicine* **2**: 187–192.

Reviewers' Comments:

Reviewer #1:

Remarks to the Author:

The authors have addressed most of my comments and questions but there is a major concern about PCoA and Bray-Curtis analyses. It is not correct as it is presented in Figure 2 – results based on non-normalized data. Normalization is necessary to equal the sequencing depth for PCoA and Bray-Curtis analyses. This can be done in different ways and one of them is to perform rarefaction to make a fair comparison between samples. In addition, why the authors deleted from M&M the R package used for PCoA and Brays-Curtis (lines 800-801) if they present their results in Figure 2?

The authors have determined NH₄, NO₃ and organic carbon contents (Supplementary Table 7) but do not discuss the nutrient stage of the soils impact on the Fe uptake and Pseudomonas.

Minor comments:

The authors have not determined the ecological interactions between plant-microbe. Ecological interactions need mathematical models. Therefore, delete "through a complex plant-microbe-plant ecological interaction" Line 36 of the Abstract.

Page 17 delete line 361 "Our findings offer novel biological insights into belowground ecological interaction networks." – this sentence misleads the readers. The authors have not determined the belowground ecological interactions.

Page 17, delete line 335-336 "Overall, our work highlights that intercropping holds high potential for a more sustainable and more ecological agriculture"

Page 757: NH₄OAc-K?

Reviewer #2:

Remarks to the Author:

The manuscript has been revised according to the suggestions made by the two reviewers. The rebuttal letter is well written and to the point (which is not always the case). For my part I was quite satisfied with the addition of the complete pyoverdine genes cluster in figure as I suggested.

Point-by-point responses to the Reviewers' Comments:

Reviewer #1:

Comment 1: The authors have addressed most of my comments and questions but there is a major concern about PCoA and Bray-Curtis analyses. It is not correct as it is presented in Figure 2 – results based on non-normalized data. Normalization is necessary to equal the sequencing depth for PCoA and Bray-Curtis analyses. This can be done in different ways and one of them is to perform rarefaction to make a fair comparison between samples. In addition, why the authors deleted from M&M the R package used for PCoA and Brays-Curtis (lines 800-801) if they present their results in Figure 2?

Response 1: Thank you for your advice. Important to note is that all results involving amplicon sequences are based on normalized data. We normalized the sequencing depth to 17,063 counts per sample using scaling with ranked subsampling (SRS) (Beule & Karlovsky, 2020). We are sorry for the incorrect response to comment 6 in the previous revision, which happened because we misunderstood your comment 6. We thought that you asked if we performed other normalization except rarefaction for PCoA and Brays-Curtis. The related information was described in lines 518-526. We choose SRS rather than the traditional rarefying method (random subsampling without replacement) because of the following reasons. 1) rarefying has poor reproducibility and can lead to distortion of the community structure. 2) SRS preserves the original community structure by minimizing subsampling errors. 3) SRS method is increasingly used in the field because of its good performance, including in some high-level articles published in *Nature Communications* (Maëva et al., 2023), *Global Change Biology* (Julia et al., 2023) and *New Phytologist* (Camille et al., 2023).

Furthermore, we did not delete the R package used for PcoA and Brays-Curtis from the methods. Instead, the related description has been moved to lines 523-526 in the Methods. In this way, the upstream and downstream analysis methods of the amplicon sequence are described together in one section, which is convenient for the reader.

Beule, L. & Karlovsky, P. (2020) Improved normalization of species count data in ecology by scaling with ranked subsampling (SRS): application to microbial communities. *PeerJ* **8**: e9593.

Camille E. D., Jessica A. M. M., Colin L., Louis J. L., Evan S. K., Randall K. K., Rodney A. C., Jason K. K., Erik A. L. (2023). Peat loss collocates with a threshold in plant-mycorrhizal associations in drained peatlands encroached by trees. *New Phytologist* **240**: 412-425.

Julia K., Cristiano B., Panos P., Arwyn J., Marc W. S., Alberto O. Maria J. I. B. (2023). Ecosystem type drives soil eukaryotic diversity and composition in Europe. *Global Change Biology* **29**: 5706-5719.

Maëva L., Cristiano B., Ferran R., Panos P., Arwyn J., Marc W. S., Vladimir M., Olesya D., Leho T., Mohammad B., Emanuele L., Marcel G. A., Alberto O. (2023). Patterns in soil microbial diversity across Europe. *Nature Communications* **14**: 3311.

Comment 2: The authors have determined NH₄, NO₃ and organic carbon contents (Supplementary Table 7) but do not discuss the nutrient stage of the soils impact on the Fe uptake and *Pseudomonas*.

Response 2: Thank you for your advice. We now elaborate on the above-mentioned issues in the discussion in lines 330-345. In brief, NH₄⁺, NO₃⁻ and organic carbon contents of soil are presented in Supplementary Table 7 and are basic physical-chemical characteristics of untreated soil used in our pot experiments. NH₄⁺ and NO₃⁻ are the main forms of nitrogen in the rhizosphere. Long-term application of NH₄⁺ decreases rhizosphere pH, while NO₃⁻ increases rhizosphere pH (Weng et al., 2022; Wang et al., 2023). Therefore, the nitrogen form can affect iron availability by altering pH. However, in our intercropping system, there is no significant pH change in the peanut rhizosphere across nutrient stages (Fig. R1, now combined with Supplementary Fig. 6), implying that the nitrogen cycle may have no direct effect on iron nutrition improvement. The reason may be that calcareous soils have strong pH buffering capacity. We further agree with the reviewer that soil properties, like NH₄⁺, NO₃⁻, or organic carbon, can potentially affect microbial community structure and function. But given that *Pseudomonas* is a generalist taxon, we do not believe that changes in these nutrients would fundamentally alter their abundance. Nonetheless, more studies relating soil properties to the abundance of keystone species for iron nutrition improvement (e.g., *Pseudomonas*) are clearly required.

Fig. R1 There is no difference in rhizosphere pH value between MP and IP, and pH remained stable over time. Points and shaded areas show mean values and SD from four independent biological replicates, respectively.

Wang Z., Tao T. T., Wang H., Chen J., Small G. E., Johnson D., Chen J. H., Zhang Y.J., Zhu Q. C., Zhang S. M., Song Y. T., Kattge J., Guo P., Sun X. (2023). Forms of nitrogen inputs regulate the intensity of soil acidification. *Global Change Biology* **29**, 4044-4055.

Weng Z. H., Li G., Sale P. W. G., Tang C. X. (2022). Application of calcium nitrate with phosphorus promotes rhizosphere alkalization in acid subsoil. *European Journal of Soil Science* **73**, e13153.

Minor comments:

Comment 3: The authors have not determined the ecological interactions between plant-microbe. Ecological interactions need mathematical models. Therefore, delete “through a complex plant-microbe-plant ecological interaction” Line 36 of the Abstract.

Response 3: Thank you very much. We agree with you that ecological interactions need mathematical models. Hence, we have deleted this sentence from the Abstract as you suggested.

Comment 4: Page 17 delete line 361” Our findings offer novel biological insights into belowground ecological interaction networks.” – this sentence misleads the readers. The authors have not determined the belowground ecological interactions.

Response 4: Thank you for your advice. We agree that this sentence could be misleading and so we have deleted it from the Discussion as you suggested.

Comment 5: delete line 335-336 “Overall, our work highlights that intercropping holds high potential for a more sustainable and more ecological agriculture”

Response 5: Thank you for your advice. We have deleted this sentence from the Discussion as you suggested.

Comment 6: NH₄OAc-K?

Response 6: The ammonium acetate (NH₄OAc) method was used to extract soil available potassium. To avoid misunderstandings, we have edited the related description in lines 762-768 in the Methods.

Reviewer #2 (Remarks to the Author):

General comment: The manuscript has been revised according to the suggestions made by the two reviewers. The rebuttal letter is well written and to the point (which is not always the case). For my part I was quite satisfied with the addition of the complete pyoverdine genes cluster in figure as I suggested.

Response: We are very grateful for this comment and your acceptance of our work. Your suggestions improved our work significantly.